# To Infinity and Beyond: Tool-Use Unlocks Length Generalization in State Space Models

**Eran Malach**
Apple
e_malach@apple.com

**Omid Saremi**
Apple
osaremi@apple.com

**Sinead Williamson**
Apple
sa_williamson@apple.com

**Arwen Bradley**
Apple
arwen_bradley@apple.com

**Aryo Lotfi**
Apple
alotfi@apple.com

**Emmanuel Abbe**
Apple
e_abbe@apple.com

**Josh Susskind**
Apple
jsusskind@apple.com

**Etai Littwin**
Apple
elittwin@apple.com

## ABSTRACT

State Space Models (SSMs) have become the leading alternative to Transformers for sequence modeling tasks. Their primary advantage is efficiency in long-context and long-form generation, enabled by fixed-size memory and linear scaling of computational complexity. We begin this work by showing a simple theoretical result stating that SSMs *cannot* accurately solve any "truly long-form" generation problem (in a sense we formally define), undermining their main competitive advantage. However, we show that this limitation can be mitigated by allowing SSMs *interactive* access to external tools. In fact, we show that given the right choice of tool access *and* problem-dependent training data, SSMs can learn to solve any tractable problem and generalize to arbitrary problem length/complexity (i.e., achieve *length generalization*). Following our theoretical finding, we demonstrate that tool-augmented SSMs achieve remarkable length generalization on a variety of arithmetic, reasoning, and coding tasks. These findings highlight SSMs as a potential efficient alternative to Transformers in interactive tool-based and agentic settings.

## 1 INTRODUCTION

Transformers (Vaswani et al., 2017), the main architecture powering large language models, have a well-known limitation: due to the attention mechanism, their computational complexity scales quadratically with the sequence length, and their memory scales linearly with length[1]. This quadratic dependency becomes a major limitation for tasks that require long-context and long-form generation. As test-time scaling paradigms that involve the generation of long Chain of Thought (CoT) (Wei et al., 2022) have become the leading solution for improving reasoning capabilities (Jaech et al., 2024; Guo et al., 2025), the ability to efficiently generate long sequences becomes even more crucial.

To solve this limitation, various works suggested replacing the attention mechanism with other modules where memory and per-token compute are fixed as a function of the sequence length (Choromanski et al., 2020). Examples of such architectures include variants of Linear Transformers (Katharopoulos et al., 2020) and State Space Models (Gu et al., 2021) such as Mamba (Gu & Dao, 2023; Dao & Gu, 2024), DeltaNet (Yang et al., 2024c) and GatedDeltaNet (Yang et al., 2024b). These architectures achieve performance similar to Transformers across a wide range of domains (Qu et al., 2024) at a lower inference cost. However, some works have also pointed out significant limitations of these architectures in certain tasks that involve memorization of long sequences and

---

[1]While a naive implementation of attention requires quadratic memory complexity, efficient implementations such as Flash Attention (Dao et al., 2022) and KV caching enable close to linear complexity.

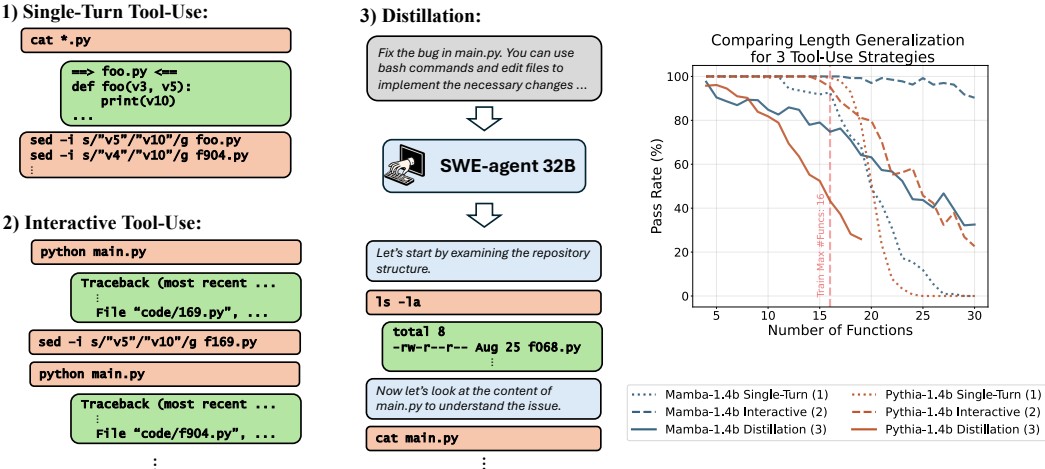

Figure 1: We finetune Mamba and Pythia (Transformer) on trajectories collected from different tool-use agents for solving a coding problem. 1) **Single-Turn Tool-Use:** Hard-coded agent that prints all the files in the repository and then outputs all the required changes. 2) **Interactive Tool-Use:** Hard-coded agent that iteratively runs the code, changes a few files, runs the code again etc. until all problems are resolved. 3) **Distillation:** SWE-agent Language Model (Yang et al., 2025) instructed to solve the bug in the code. Models are trained on codebases of up to 16 files (dashed red line), with context length 8K, and evaluated on larger codebases with longer context. While Pythia outperforms Mamba on smaller codebases and single-turn tool use, Mamba displays favorable performance on large codebases when trained to imitate interactive agents (agents 2 and 3), extrapolating beyond the training distribution.

in-context learning (Jelassi et al., 2024; Park et al., 2024; Akyürek et al., 2024). Possibly due to these limitations, linear-time models are still not widely adopted as a replacement to Transformers.

The goal of this work is to understand the capabilities and limitations of SSMs, focusing on tasks that require long-form generation. We formally define long-form generation tasks to be problems where the effective number of *outputs* grows with the complexity of the problem. We focus on such tasks as these are the tasks where SSMs display a clear benefit over Transformers in terms of inference efficiency. However, we show that this efficiency comes at a cost of inherent performance degradation. Namely, we prove that SSMs *fail* to solve long-form generation tasks when the complexity of the task increases beyond the capacity of the model, even if the model is allowed to generate CoT of any length. This limitation arises from the bounded memory of the model, which limits the expressive power when generating long sequences. This is in contrast with Transformers which, using CoT, can in principle solve any computationally tractable problem, utilizing their unbounded memory (Merrill & Sabharwal, 2023). So, to solve long-form generation tasks we can either use Transformers and suffer quadratic scaling of compute, or use SSMs and suffer performance degradation. Another alternative is to use hybrid models that mix attention and SSM layers and have been recently shown to achieve state-of-the-art performance at large scale (Blakeman et al., 2025). However, this ultimately does not eliminate the quadratic dependence on the sequence length.

Following the observation above, we explore another alternative: allowing SSMs to *interactively* use external tools. LLMs are now increasingly used as agents that interact with external tools for solving tasks such as coding, math or question answering (Luo et al., 2025; Yehudai et al., 2025). These tools can allow agents to query and read from external resources, and write information that can be used later. Therefore, such tool-use can naturally augment the internal memory of the model, allowing it access to practically unbounded external memory. We introduce a new theoretical framework for studying ReAct (Yao et al., 2023) agents, and show that allowing SSMs access to external memory through interactive tool-use makes them much more powerful. We prove that tool-augmented SSMs trained on task-specific trajectories can achieve *length generalization* on any tractable long-form generation task. That is, we show that for any such task we can construct training data with tool-use trajectories such that a simple training paradigm learns to execute the task with high accuracy, even when evaluated beyond the length of the training data. Importantly, this result only holds for *interactive* tool-use, and we show that *single-turn* tool-use SSMs are still limited.

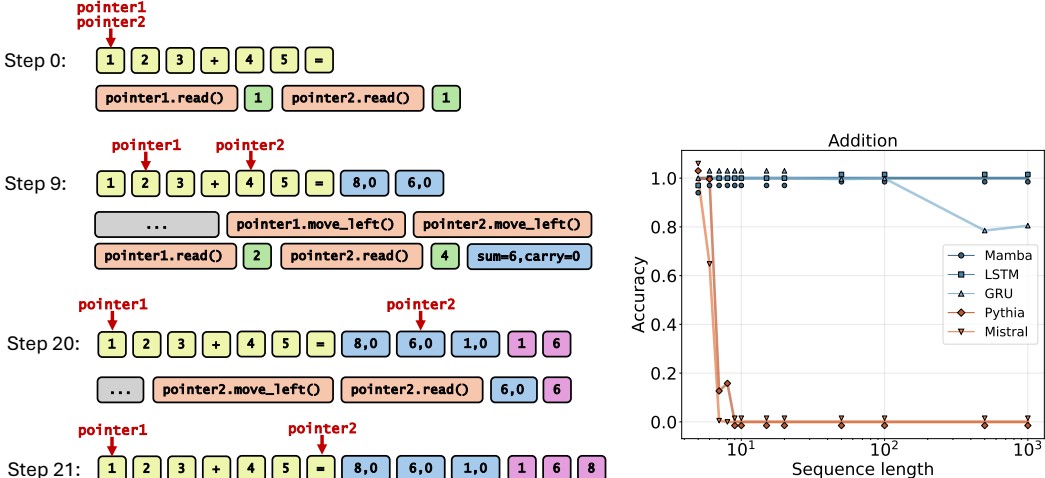

Figure 2: **Left:** Illustration of an interactive tool-use agent trajectory with pointer-based memory tool for solving multi-digit addition. The agent can generate *thoughts* (blue), *outputs* (purple) or *commands* (orange), and receive *observations* (green) from the memory tool. At each step, we show the state of the memory context on the top row, and below it show the sequence of generated tokens. **Right:** Accuracy of recurrent/SSM models (Mamba, LSTM, GRU) and Transformers (Pythia, Mistral) trained on trajectories for $\leq$ 5-digit addition, evaluated on up to 1,000-digits (log scale).

Experimentally, we show that SSMs trained to *interactively* use external memory tools achieve length generalization on tasks such as arithmetic, logical reasoning and coding. For example, a Mamba model trained to solve a simple coding task extrapolates to codebases larger than those seen during training when trained on trajectories with interactive tool-use (Figure 1). Additionally, a Mamba model trained to execute long-form multi-digit addition using pointer-based memory can generalize from 5-digit addition to 1,000-digit addition (Figure 2). We observe similar results on multiplication and on a logical reasoning task, and more modest extrapolation on solving the Tower of Hanoi task (a task which proved to be difficult for reasoning models, see Shojaee et al. (2025)). Taken together, our theoretical and experimental results highlight the potential advantage of using SSMs as agents with interactive tool access, instead of using them as standalone systems.

## 1.1 RELATED WORK

**Chain-of-Thought and Scratchpad**   When solving problems that require reasoning, LLMs are known to significantly benefit from generating a CoT, detailing the step-by-step process required for solving the target task (Wei et al., 2022; Nye et al., 2021). Indeed, many datasets used for training models on mathematical problems include such CoT in the training data (Toshniwal et al., 2024b;a; Cobbe et al., 2021). Theoretically, CoT is shown to improve both the expressive power of language models (Merrill & Sabharwal, 2023) and their optimization and learnability (Wies et al., 2022; Malach, 2023). Additionally, it was shown that choices of CoT training data that "localize" the computation enable efficient learning and length generalization (Abbe et al., 2024). In another work, using CoT that encodes the operation of a Turing machine was used to improve length generalization on various tasks (Hou et al., 2024; Hua et al., 2025). In our work, we follow a similar approach for improving length generalization capabilities of language models. However, we focus on SSMs instead of Transformers, and study the effect of interactive tool-use for improving learning and generalization.

**Emulations and Neural Turing Machines**   The goal of learning to execute general algorithms with neural networks has been discussed in various works. Abbe & Sandon (2023) and Abbe et al. (2021) show universality learning properties of poly-size neural networks trained by stochastic gradient descent. Graves et al. (2014) introduces the Neural Turing Machine (NTM), a neural network that can simulate Turing machines and thus execute computable algorithms. NTMs were studied in different settings (Malekmohamadi Faradonbe et al., 2020), with some improvements such as the Neural GPU (Kaiser & Sutskever, 2015), but were ultimately not widely adopted. Similar works suggested augmenting LSTMs (Hochreiter & Schmidhuber, 1997) with external stack or external tape (Delétang et al., 2022; Joulin & Mikolov, 2015). We use similar ideas to study algorithmic

learning and length generalization capabilities of SSMs in the setting of tool-augmented interactive agents.

**Length Generalization** The problem of length generalization, training models on short/simple problems and evaluating them on longer/complex instances, has been studied in many works. These works often focus on training Transformers on arithmetic or algorithmic tasks such as sorting, copying or multi-digit addition (Jelassi et al., 2023; Nogueira et al., 2021). Different works suggest various techniques for improving length generalization capabilities of Transformers, including various choices of positional encodings and output format (Zhou et al., 2024; Cho et al., 2024; McLeish et al., 2024; Kazemnejad et al., 2023; Ruoss et al., 2023), scratchpads (Nye et al., 2021; Lee et al., 2023; Zhou et al., 2023; Abbe et al., 2024), architecture (Ontanon et al., 2021; Li & McClelland, 2023), mixing different tasks for "task hinting" (Awasthi & Gupta, 2023) or using looped Transformers (Fan et al., 2024). Some works aim to give scientific or theoretical explanation to the capability and limitation of Transformers in extrapolating beyond the context length (Golowich et al., 2025; Zhou et al., 2023; Huang et al., 2024; Bhattamishra et al., 2022). SSMs have been shown to display robust length generalization capabilities in certain cases. Gu & Dao (2023) demonstrate that Mamba achieves significantly better length generalization performance compared to Transformers on some tasks. Other works show that the length extrapolation of SSMs can be significantly improved with modifications to the model (Ben-Kish et al., 2024) or the training pipeline (Ruiz & Gu, 2025). In this paper, we study the length generalization of SSMs when trained on data with tool-use trajectories. We show that SSMs can achieve perfect length generalization in this setting on various tasks.

## 2 THEORY

In this section we formally define the notion of long-form generation tasks: tasks that require generating longer output sequences as their complexity increases. Following this, we define a family of functions that generalizes the class of SSMs, and theoretically analyze their limitation and capabilities in different tool-use settings.

**Definitions and Notation.** Fix some set $\mathcal{Z}$ and some distribution $\mathcal{P}$ over $\mathcal{Z}$. For some subset $S \subseteq \mathcal{Z}$, we denote by $\mathcal{P}(S)$ the probability mass of $S$ under $\mathcal{P}$, i.e. $\mathcal{P}(S) := \Pr_{z \sim \mathcal{P}}[z \in S]$. For some function $f : \mathcal{Z} \to \mathcal{Z}'$, we denote by $f(\mathcal{P})$ the probability distribution of $f(z)$ for $z \sim \mathcal{P}$. For some set $B$, we denote by $\Delta(B)$ the set of probability distributions over $B$.

**Definition 2.1.** *For some finite set $\mathcal{Z}$ and some distribution $\mathcal{P}$ over $\mathcal{Z}$, we define the **minimum support size** of mass $0 \leq \alpha \leq 1$ for $\mathcal{P}$ to be the size of the smallest set that covers $\alpha$ probability mass:* $\mathrm{supp}_\alpha(\mathcal{P}) := \min\{|S| : S \subseteq \mathcal{Z}, \mathcal{P}(S) \geq \alpha\}$.

### 2.1 LONG-FORM GENERATION

Let $\Sigma$ be a dictionary of tokens, and denote by $\Sigma^*$ the set of strings of tokens. Let $\mathcal{X}_1, \mathcal{X}_2, \cdots \subseteq \Sigma^*$ be a sequence of input spaces, and let $\mathcal{Y}_1, \mathcal{Y}_2, \cdots \subseteq \Sigma^*$ be a sequence of output spaces. We assume that the input and output spaces are finite, i.e. $|\mathcal{X}_n|, |\mathcal{Y}_n| < \infty$ for all $n$. Let $\mathcal{D}_1, \mathcal{D}_2, \ldots$ be a sequence of distributions, such that $\mathcal{D}_n$ is a distribution over $\mathcal{X}_n$. Finally, let $f : \Sigma^* \to \Sigma^*$ be some ground-truth function that satisfies, for all $n$, that $f(\mathcal{X}_n) \subseteq \mathcal{Y}_n$. We think of the parameter $n$ as a complexity parameter, and so the distribution $\mathcal{D}_n$ generates more complex inputs as $n \to \infty$. We give the following definition of long-form generation tasks:

**Definition 2.2.** *We say that $f, \{\mathcal{D}_n\}_{n=1}^\infty$ is a **long-form generation** task with coverage $\alpha \in (0,1)$ if* $\mathrm{supp}_\alpha(f(\mathcal{D}_n))$ *is monotonically increasing with $n$,[2] and $\lim_{n\to\infty} \mathrm{supp}_\alpha(f(\mathcal{D}_n)) = \infty$.*

Namely, we require that as the complexity $n$ increases, the effective number of possible outputs (i.e., outputs that have non-negligible probability mass) increases as well. We note that many natural long-form generation tasks indeed satisfy these conditions, for example: 1) **Multi-Digit Addition** (or Multiplication): $\mathcal{D}_n$ is a distribution over strings of the form $a + b =$ (or $a \times b =$), where $a, b$ uniformly random numbers with $n$-digits. The function $f$ maps the input strings to the solution, e.g. $f(\text{``}a + b = \text{''}) = c$ where $c = a + b$ (or $c = a \cdot b$). 2) **Sorting**: $\mathcal{D}_n$ is a distribution over $n$ items,

---

[2]The condition that the support size is monotonically increasing makes the theoretical results slightly easier to introduce, and holds for practically all reasonable long-form generation problems.

$f$ maps to the sorted list of items. 3) **Code Fixing**: $\mathcal{D}_n$ is a distribution over python codes that have bugs that require changing $n$ lines of code, $f$ maps the code to the necessary changes.

## 2.2 Generalized State Space Models

In this section, we follow similar definitions and notations as in Jelassi et al. (2024). We define a state space to be some *finite* set $\mathcal{S}$ with $|\mathcal{S}| < \infty$. A generalized state space model (GSSM) is a (potentially probabilistic) function $h : \Sigma^* \to \Delta(\Sigma^*)$ defined by an initial state $s_0 \in \mathcal{S}$ and two rules: an update rule $u : \mathcal{S} \times \Sigma \to \mathcal{S}$, and an output rule $r : \mathcal{S} \to \Delta(\Sigma)$. Given some input $\boldsymbol{x} \in \Sigma^*$, the function $h$ generates a sequence $\boldsymbol{y} \in \Sigma^*$. We define the state and the output of $h$ at time $t$ recursively s.t. $s_t = u_h(s_{t-1}, x_{t-1})$ if $t \le |\boldsymbol{x}|$ and $s_t = u_h(s_{t-1}, y_{t-|\boldsymbol{x}|})$ if $t > |\boldsymbol{x}|$, and we sample $y_t \sim r(s_{|\boldsymbol{x}|+t})$. We terminate when an end-of-sequence token $[\text{EOS}] \in \Sigma$ is observed.

Note that any model that has fixed memory as a function of the sequence length satisfies the definition of a GSSM. This includes common choices of recurrent models, such as: LSTM (Hochreiter & Schmidhuber, 1997), Linear Transformers (Katharopoulos et al., 2020), H3 (Fu et al., 2022), RetNet (Sun et al., 2023), Mamba-1 and Mamba-2 (Gu & Dao, 2023; Dao & Gu, 2024) and other variants (Yang et al., 2024b). Additionally, Transformers where all the attention layers have local (sliding window) attention with fixed window size, are also GSSMs. Other computational models that use Transformers to process fixed length sequences and update fixed-size "memory" vectors (Hutchins et al., 2022) are also GSSMs. Transformers and hybrid-SSM models are *not* GSSMs, since their memory increases with the sequence length.

**CoT, Single-Turn and Interactive Tool-Use.** We analyze multiple settings where the model can invoke reasoning and tool-use. We follow the popular ReAct framework (Yao et al., 2023) and let the model generate either *thoughts*, that capture the internal reasoning of the model, or *actions* that are followed by *observations* from the environment. The thoughts and actions can be interleaved during the runtime of the model. We specify two types of actions: *command actions*, that are sent to a tool-oracle $\mathcal{O}$ that returns an *observation* following the execution of the command, and *output actions* that are simply tokens appended to an *output stream* and do not result in an observation. The *output stream* captures the final response of the model which is then evaluated against the ground-truth[3]. Thoughts, commands and observations are placed between dedicated open/close tags (e.g. $[\text{THINK}], [\backslash\text{THINK}]$). We define more formally the tool-oracle and the interaction protocol in Appendix A. We analyze three settings for problem-solving agents. 1) **CoT-only:** The model is allowed to use only *thoughts* or *outputs*, but cannot issue *commands* or receive external observations[4]. 2) **Single-Turn Tool-Use:** The model is allowed to issue a single *command*, followed by an observation, and then generate the output. The model can use *thoughts* before and after the tool call, and during the output generation. 3) **Interactive Tool-Use:** The model is allowed to use as many *commands* as it needs, and freely interleave thoughts, commands and outputs.

## 2.3 Learning Algorithms and Length Generalization

Fix some task $f, \{\mathcal{D}_n\}_{n=1}^\infty$. We now define training data distributions for learning the task $f$. We note that for many downstream tasks, it is common to collect training data that contains CoT reasoning and/or tool-use traces for solving the problem. We therefore allow the training distributions to contain a task-specific reasoning and tool-use trajectories. Given some trajectory $\boldsymbol{z} \in \Sigma^*$, we denote by $\boldsymbol{z}^{(\text{out})}$ the value of the output stream after execution of the trajectory. Formally, a training distribution for the task $f, \{\mathcal{D}_n\}_{n=1}^\infty$ is a sequence of distributions $\{\mathcal{P}_n\}_{n=1}^\infty$ s.t. $\mathcal{P}_n$ is a distribution over $\mathcal{X}_n \times \Sigma^*$ satisfying that: 1) $\mathcal{D}_n$ is the marginal distribution of $\mathcal{P}_n$ w.r.t. $\mathcal{X}_n$, and 2) For $(\boldsymbol{x}, \boldsymbol{z}) \sim \mathcal{P}_n$, with probability 1 it holds that $\boldsymbol{z}^{(\text{out})} = f(\boldsymbol{x})$ (i.e. the output stream at the end of generation evaluates to the correct answer).

A learning algorithm $\mathcal{A}$ is an algorithm that, for some given length $n$, draws a sample of size $m$ from $\mathcal{P}_1, \ldots, \mathcal{P}_n$[5], and returns some hypothesis $h : \Sigma^* \to \Delta(\Sigma^*)$ that, given an input problem, can gen-

---

[3]We focus on agents for solving input-output problems, where the task of the model is to generate output given the input problem (e.g. question answering, coding, mathematical proofs etc.). This is a different setting from an agent that performs actions and collects rewards, as in many Reinforcement Learning problems.

[4]This setting also includes the case where the model generates the output immediately, without using CoT.

[5]We let the algorithm choose freely how to sample from these distributions.

erate a reasoning and tool-use trajectory. We denote the output of $\mathcal{A}$ in this case by $\mathcal{A}(\mathcal{P}_1, \ldots, \mathcal{P}_n)$. We say that $\mathcal{A}$ is a GSSM learning algorithm if it always returns a GSSM. We define the error of $h$ w.r.t $f$ for some complexity $n$ by $\text{err}_n(h) = \Pr\left[h^{(\text{out})}(\boldsymbol{x}) \neq f(\boldsymbol{x})\right]$, with probability over $\boldsymbol{x} \sim \mathcal{D}_n$ and the randomness of $h$. We now define *length generalization* of an algorithm:

**Definition 2.3.** *We say that $\mathcal{A}$ achieves **length generalization**, if for every $\epsilon, \delta \in (0,1)$ there exists some minimal complexity $n_0$ and sample size $m$ s.t. w.p. $\geq 1 - \delta$ we have that $h_{n_0} = \mathcal{A}(\mathcal{P}_1, \ldots, \mathcal{P}_{n_0})$ satisfies $\text{err}_n(h_{n_0}) \leq \epsilon$ for all $n \geq n_0$.*

Namely, we require that the algorithm returns a hypothesis with low-error on problems with arbitrarily large complexity $n$, as long as it sees "complex enough" inputs sequence in the training data (with complexity larger than $n_0$). This requirement may seem relatively strong, as we could expect that the error of the learned model would grow with the complexity of the problem. However, we will show theoretically (and to some extent, empirically) that with a carefully constructed training data, achieving such "infinite" length generalization is possible.

## 2.4 MAIN RESULTS

In this subsection, we state the main theoretical results in the paper. We begin by showing a negative result, stating that GSSMs *cannot* solve long-form generation tasks, if they operate in the *CoT-only* or *single-turn tool-use* setting. Following this, we show a positive result, proving that for any computable long-form generation task we can construct training data such that a simple learning algorithm achieves length generalization on the target task in the *interactive tool-use* setting.

**GSSMs cannot Solve Long-Form Generation Tasks without Interaction.** We begin by stating the negative result. The proof is relatively simple: since the model has a fixed memory, and outputs are a function of the state of the memory, the model cannot generate all outputs as complexity grows.

**Theorem 2.1.** *Let $f$ be a long-form generation task over $\{\mathcal{D}_n\}_{n=1}^{\infty}$ with coverage parameter $\alpha \in (0,1)$. Then, for any CoT-only or Single-Turn GSSM $h$ there exists some problem complexity $n_0$ s.t. for all $n \geq n_0$ the model $h$ has error: $\text{err}_n(h) \geq 1 - \alpha$.*

The full proof is given in Appendix B. An immediate implication of this result is that GSSM learning algorithms cannot achieve length generalization on long-form generation tasks without interaction.

**GSSMs with Interactive Tool-Use can Length Generalize on Long-Form Generation Tasks.** For some function $f : \Sigma^* \to \Sigma^*$, we say that $f$ is *computationally tractable* if there exists a Turing machine $\mathcal{T}$ s.t. for any $\boldsymbol{x} \in \Sigma^*$, if $\mathcal{T}$ begins with $\boldsymbol{x}$ written on its tape, it halts with $f(\boldsymbol{x})$ written on its tape. The following result shows that a GSSM learning algorithm can achieve length generalization with interactive tool-use, given proper training data:

**Theorem 2.2.** *There exists memory-tool oracle $\mathcal{O}$ and a simple GSSM learning algorithm $\mathcal{A}$[6] s.t. for any computationally tractable long-form generation task $f, \{\mathcal{D}_n\}_{n=1}^{\infty}$, there exists a sequence of training distributions $\{\mathcal{P}_n\}_{n=1}^{\infty}$ for which $\mathcal{A}$ achieves length generalization in the interactive setting, with sample complexity $m = n_0 M \log(M/\delta)/\epsilon$ (where $M$ is a constant that depends on the Turing machine for computing the function $f$).*

To show the above result, we define a simple tool that allows read/write access to some external memory, using a pointer that can move left or right between the memory cells. Using this tool, we can simulate the operations of a Turing machine, where we use the external memory as the tape of the Turing machine, use *thoughts* to track the state of the machine and *commands* to move the head and read/write symbols. Since the transition function of the Turing machine is defined for every pair of state and symbol, to prove that length generalization is achieved we show that, for large enough $n_0$, most of these pairs are seen in the training data. We give the complete proof in Appendix B.

To conclude, the above results show that *interactive* tool-use is both *necessary* and *sufficient* for GSSMs to achieve length generalization on tractable long-form generation problems.

---

[6] The algorithm that we analyze is "simple" in the sense that it learns a function that operates using simple string-matching with the training data, similar to e.g. n-gram models. While this is not a "standard" learning algorithm, we believe that similar results can be obtained for more natural algorithms (e.g., gradient descent on a simple RNN), at the cost of making the analysis much more involved.

Table 1: Experimental results for synthetic tasks for different models. The notation $n \to m(p\%)$ means a model trained on length $n$ achieves accuracy $p$ on length $m$ (for the largest $m$ s.t. $p \geq 5\%$).

| Model | $n \times 1$ | $n \times 2$ | Logical Graph | Hanoi[7] |
|---|---|---|---|---|
| Mamba | **10→1K (100%)** | **10→1K (100%)** | 10→1K (98%) | **8→12 (49%)** |
| LSTM | 10→500 (100%) | 10→100 (100%) | **10→1K (100%)** | 8→8 (100%) |
| GRU | 10→500 (100%) | 10→100 (100%) | **10→1K (100%)** | 8→8 (100%) |
| Pythia | 10→20 (79%) | 10→14 (12%) | 10→1K (5%) | 8→8 (100%) |
| Mistral | 10→13 (25%) | 10→20 (33%) | 10→500 (9%) | 8→8 (100%) |

## 3 EXPERIMENTS

In this section we evaluate the length generalization capabilities of GSSMs and Transformer-based language models on various tasks, including arithmetic, reasoning and coding. We experiment with different choices of tools that allow read/write memory access, using either a pointer-based memory access, search tool, or arbitrary *bash* commands for reading and changing files. We use both tasks where we synthetically generate the ground-truth trajectory and tool commands, as well as a coding task where we collect the trajectories from a SWE coding agent. In our experiments, we largely follow a similar framework for ReAct agents defined in the previous section, where the model can interleave *thoughts*, *outputs* ("final answer" tokens), and *commands* that are followed by *observations* from the environment. In our experiments we use Mamba SSM (Gu & Dao, 2023), LSTM (Hochreiter & Schmidhuber, 1997), GRU (Cho et al., 2014), Pythia Transformer (Biderman et al., 2023) and a Transformer with sliding window (local) attention based on the Mistral architecture (Jiang et al., 2023). In all experiments, we see that SSMs/RNNs achieve length generalization performance that is much better compared to Transformers. See Appendix C for experimental details.

### 3.1 ARITHMETIC TASKS

In the following set of experiments, we augment the model with a pointer-based memory tool that gives the model access to past tokens in the input/output context. In this setting, the model can execute the following commands: 1) initialize a new pointer, 2) move the pointer left or right by a single token and 3) read the token under a given pointer. By default, a new pointer is initialized to the first token position of the input context. The *thoughts* and *outputs* are appended to the context, and are therefore accessible by the pointers (if they reach beyond the length of the input), but *commands* and *observations* are discarded (i.e., they are not appended to the context memory and cannot be read by the pointers). We give a detailed description of how *thoughts*, *commands* and *outputs* are specified in Appendix D.1. The final answer is written in the *output stream* at the end of the generation.

We train the model using the standard next-token prediction objective with teacher-forcing, while masking from the loss the input question and the *observations* (the outputs of a *read* operation, which will be generated by the memory tool). For the training data, we construct synthetic trajectories that simulate the desired algorithm, and train the model to exactly execute the algorithm required for solving the problem using the memory-tool interaction.

**Multi-Digit Addition.** For this task, we train the model to perform multi-digit addition. We fix some maximal training length $n$, and for each training example we sample uniformly $n_1, n_2 \sim \{1, \ldots, n\}$, then sample two numbers $x_1, x_2$ where $x_i$ is a uniformly random $n_i$-digit number. We construct a training example with the trajectory for solving $x_1 + x_2$, essentially mimicking the long addition algorithm (see Appendix D.3 for details). For evaluation, we choose $n' \geq n$ and evaluate on addition of two $n'$-digit numbers. In evaluation, we measure the accuracy of exact-recovery of the trajectory *and* the final answer (i.e., we measure the probability of generating a solution that exactly matches the desired algorithm). Figure 2 (right) shows the results of this experiment. We observe that Mamba and LSTM trained on 5-digit demonstrations learn to perfectly perform 1,000-digit addition (we did not measure the accuracy beyond this). A Transformer trained in the same setting fails to extrapolate. Additional ablations, such as training with no CoT, no tool-use

---

[7]Due to the sensitivity of the Hanoi experiments to the initialization seed, we use 10 seeds for Mamba and Pythia and 3 seeds for other models. Best performing seed is reported. See Appendix D.4 for more details.

and single-turn tool-use, result in little to no length generalization, and are discussed in Appendix D.7. We also compare Mamba to two alternative architectures on the addition task: Hybrid-Mamba, which interleaves Mamba layers with attention layers, and Recurrent Memory Transformer (RMT) (Bulatov et al., 2022), which augments a Transformer with learnable memory tokens and segment-level recurrence. We find that Hybrid-Mamba performs similarly to Mamba, while RMT does not achieve meaningful length generalization. See Appendix D.9 for details.

**Multi-Digit Multiplication.**    For this task we use the same pointer-based memory tool described above for learning the multiplication algorithm. In this task, we increase the length of only the first operand, keeping the second operand fixed. Specifically, we fix some maximal training length $n$, choose $n_1 \sim \{1, \ldots, n\}$ to be the length of the first operand and choose $n_2 \sim \{1, 2\}$ to be the length of the second operand (i.e. we multiply an $n_1$-digit number by a 1-digit or 2-digit number). We sample $x_1, x_2$ where $x_i$ is a uniformly random $n_i$-digit number, and construct the trajectory for solving $x_1 \times x_2$ (see details in Appendix D.3). We then evaluate on $n' \times 1$ and $n' \times 2$ digit multiplication, for some $n' \geq n$, and report exact recovery accuracy. We train different SSMs/RNNs and Transformers where first operand has $n \leq 10$ digits, and evaluate on multiplications of numbers with up to 1,000-digits (Table 1). Here too we see that Mamba models maintain high accuracy when evaluated on numbers that have orders of magnitude more digits than in the training data (also see Appendix D.6 for ablations on training steps and maximum number of digits seen during training).

**Task Mixture.**    We examine whether co-training a primary task with an auxiliary task that shares a related computational structure yields synergistic benefits (Awasthi & Gupta, 2023). Our experiments indicate that such co-training improves the length generalization of the primary task under limited training budgets. In our experiments, the primary task is multiplication ($n$-digit × 2-digit), co-trained with addition ($n + n$ digits) as an auxiliary task. Both tasks share structural similarities when expressed as sequences of tool calls. The training distribution for multiplication contains samples up to 20 digits. We compare the accuracy as a function of test length for various training budgets (250, 500 or 800 steps) and various choices of task mixtures (see Appendix D.8). We observe that under limited budgets (250 steps), introducing auxiliary addition samples yields minor improvements. At intermediate budgets (500 steps), the benefit becomes more pronounced, with certain weights extending generalization to much larger $n$. However, with sufficient training (800 steps), all settings converge to strong generalization.

## 3.2    ALGORITHMIC/REASONING TASKS

We next turn to evaluate the tool-use paradigm on tasks that test certain "reasoning" capabilities.

**Tower of Hanoi.**    This task is based on the popular puzzle game, which was also recently used for testing reasoning capabilities of frontier LLMs, showing that they struggle to solve this task as complexity increases (Shojaee et al., 2025). In our setup, we randomly sample (without replacement) $n$ disks of sizes $\in \{1, \ldots, 100\}$. These disks are placed on the first rod (labeled $A$), ordered from the largest to the smallest, with rods $B$ and $C$ being empty. The input to the model is the list of disks, which captures the initial state of the game. The model then needs to output a sequence of valid moves that result in placing the pile on rod $C$. We use the same pointer-based memory tool as in the previous experiments, and train the model trajectories with up to $n$ disks, evaluating on larger $n'$ (see Appendix D.3). In this experiment we observe more limited length generalization (Table 1), but note that unlike other experiments, here the length of the output increases *exponentially* with $n$.

**Logical Graph.**    In this task, we construct a directed-acyclic computation graph with $n$ nodes. The graph has $k$ input nodes (for some fixed $k$), and each internal node node computes a Boolean operation (AND/OR) on one or two input variables or their negations. We construct the graph by iteratively adding new internal nodes and randomly choosing their Boolean operation and their connectivity to existing nodes in the graph. We take the last node that is added to be the output node. All nodes are randomly labeled, and the model receives the graph structure and an assignment for the input variables as a python code (see Figure 4). In this task, instead of using the pointer-based memory tool as in previous tasks, we use a *search* tool: the model can issue a command $\texttt{find}(x)$, and gets a list of all occurrences of the pattern $x$ in the context. As before, all *thoughts* and *outputs* generated by the model are appended to the context and are therefore searchable in future iterations.

We fix $k = 3$ and train the model on trajectories for solving this problem for graphs with $n \leq 10$ nodes. We evaluate on graphs with $n' \geq n$ nodes, and report the exact-match accuracy in Table 1. We observe that Mamba and recurrent models extrapolate to graphs with $n = 1K$ nodes.

## 3.3 Coding Task

For the previous tasks, we trained models "from scratch" on synthetic trajectories that invoke tool use for solving arithmetic and algorithmic problems. This allowed us to demonstrate the length generalization capability of SSMs equipped with tool-use in a clean and controlled setting, resulting in perfect recovery of the underlying algorithm in many cases. We now turn to study extrapolation of tool-use agents in a more realistic coding setting. Importantly, this setting will allow us to go beyond programmatically generated trajectories and collect trajectories from an existing SWE coding agent. This demonstrates that our results and observations can also be applicable in settings where the underlying algorithm/method for solving the task are not known or well-specified.

Our task will be fixing a "bug" in a given codebase. To construct the codebase we generate $n$ python functions, each function saved in a separate python file. The functions form a dependency graph, with one root function called `main` (stored in `main.py`). Each function declares variables (named `v0,v1,...,v9`), gets some variables as inputs and passes some variables to other functions it imports. We generate this codebase by randomly generating a dependency graph, iteratively adding nodes (functions) to this graph and connecting each node to existing nodes, where each edge represents a parent function importing a child function. Function names are randomly selected from `f0,...,f999`, except for the last function added to the graph which is called `foo`. We then randomly assign variables and print them and/or pass them from parent functions to child functions. The code always has the following "bug": there is a special variable `v10` that is declared in `main.py` and is called in `foo.py` without properly passing it from `main`. In order to fix the code, we need to pass the variable `v10` through all the dependency paths from `main` to `foo` (ideally without changing other functions, though we do not enforce this). See Figure 4 for an illustration.

We start by running a coding agent and collecting its trajectories when attempting to solve this code-fixing task, as we are varying the number of functions $n$ in the codebase (choosing $n \in \{4, \ldots, 16\}$). We use three types of agents for generating trajectories, (illustrated in Figure 1): 1) **Single-Turn Agent**: Hard-coded agent that prints all the files and immediately generates the correct code edits. 2) **Interactive Agent**: Hard-coded agent that iteratively runs the code, resolves the issue in up to 3 files, then runs the code again, and keeps going until the code runs without errors. 3) **Distillation**: An agent based on `SWE-agent-LM-32B` (Yang et al., 2025), a capable open-source coding model that we couple with mini-SWE-agent[8] (Yang et al., 2024a) as a simple agent environment which gives the model access to the code through bash commands. We instruct the model to fix the bug in the code, specifically telling it what the bug is and how it should fix it (pass the variable `v10` from `main` to `foo`). See the full prompt and further details in Appendix E. We observe that while this task is relatively simple, the model's performance degrades as the complexity (number of functions) in the codebase increases (see statistics in Appendix E). We therefore filter the trajectories to include only trajectories that correctly fixed the code, and also filter for short trajectories (shorter than the average length for a given size $n$).

After collecting around 100K trajectories from each coding agent, we finetune two comparable models on these trajectories: Pythia-1.4B (Transformer-based model, Biderman et al. (2023)) and Mamba-1.4B (Gu & Dao, 2023), both pretrained on The Pile (Gao et al., 2020). We train both models with context length 8,192, on codebases of up to 16 functions (if the trajectory is longer than the context length, we train only on the first 8,192 tokens). We then evaluate both models on codebases of different sizes, letting the models generate beyond the context length.[9] We measure the probability of correctly fixing the code (using the same environment used for collecting the trajectories). As shown in Figure 1, we observe that for codebases with small number of functions, both Transformer and Mamba models perform well in all settings. Notably, the Transformer-based model outperforms the Mamba SSM for small codebases in the agent distillation setting, achieving over 90% pass rate. However, for larger codebases, beyond the training distribution (both in terms

---

[8] https://github.com/SWE-agent/mini-swe-agent

[9] We experimented with applying RoPE scaling when using the Transformer beyond the training context length, both in finetuning and evaluation, and observed mixed results. We report the accuracy for the best choice (with or without RoPE scaling) in each setting.

of number of functions and trajectory length), we see that the Mamba model maintains much better accuracy as the complexity increases when trained to imitate interactive agents (agents 2 and 3), but fails on complex codebases when trained in the single-turn setting (agent 1). This finding aligns with our theoretical results, and also matches the previous synthetic experiments.

### 3.4 LONG-CONTEXT NATRUAL-LANGUAGE

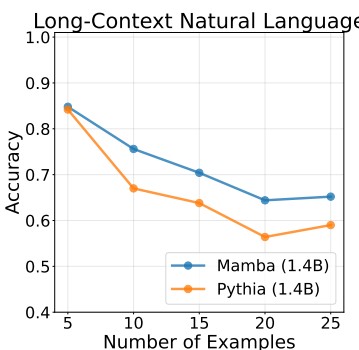

To further compare the length generalization of SSMs and Transformers in long-context tool-use setting, we train our models on solving a task that involves natural language and long-context reasoning. In particular, we adapt a task from the Oolong benchmark (Bertsch et al., 2025), a very recent benchmark that proposed a novel set of long-context tasks that prove to be hard for frontier models. We choose a representative task from this benchmark, in which the model is presented with a dataset of natural-language text examples[10], each with a particular date, and needs to answer questions of the form: *"In the above data, was a question of category X more common, less common, or the same frequency before date Y, as compared to after date Y?"*. We train the models on hard-coded agent trajectories that use a simple tool call that retrieves the next example in the dataset (starting from the first example), along with *thought* tokens for analyzing each example. We finetune 1.4B Pyhita Transformer and Mamba SSM (similarly to the coding experiment) on problems with up to 5 data examples, and compare the performance of the models when tested on larger number of examples. As shown in Figure 3, here too we observe that while both models have similar performance in-distribution, the Mamba SSM extrapolates better when evaluated on longer datasets.

Figure 3: Accuracy of Pythia and Mamba trained on in-context datasets with 5 examples, evaluated on up to 25 examples.

## 4 CONCLUSION AND DISCUSSION

We started this work by comparing two families of models for long-form generation: Transformers and SSMs. Transformers are inefficient for long-context and long-form generation, as their computational complexity scales quadratically with the sequence length. SSMs, on the other hand, offer linear scaling of compute but, as we showed, cannot accurately solve long-form generation tasks (without tools). This demonstrates a clear trade-off between efficiency and accuracy that seems to be inescapable. Indeed, several works have observed that SSMs are inferior to Transformers in various tasks that require memorization of long sequences (Jelassi et al. (2024); Waleffe et al. (2024)).

On the positive side, we show that in the agentic/tool-use setting, SSMs can leverage tools to overcome their memory bottleneck, thus offering efficiency, accuracy, and generalization to longer sequences. In hindsight, SSMs seem to be a natural fit for tool-use settings: tools often generate large quantities of content, which SSMs can parse efficiently, and also involve multi-turn interactions that can quickly overflow the context of a standard Transformer. However, it seems that there is little work on building SSM-based agents, and thus their evaluation is restricted to the "standalone" setting, where they are inherently limited. We do not believe this is due to any inability of SSMs to learn tool-use behavior. For example, while Mistral's Mamba-Codestral-7B-v0.1 model does not naively support function calling, it is able to achieve comparable tool-use performance to several function-calling-enabled transformer-based models (Appendix F). We therefore believe this work should encourage the development of a tool-based SSMs that operate in various agentic settings, such as coding, search or reasoning. This application could potentially unlock the full capabilities of these models, making them competitive with, or even superior to, Transformer-based agents.

Finally, we want to emphasize that our work is one of the first to analyze the performance of language modeling architectures *in a system*, rather than as standalone models. To some extent, our analysis shows that certain architectures can be "weaker" when operating standalone, but in fact perform better when incorporated as part of a system. Since LLMs are now rarely used as standalone tools, we believe that this aspect of language modeling deserves more attention and focus in the field.

---

[10]We use examples from the Yahoo Answers Topic Classification dataset (Zhang et al., 2015).

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

# A MORE DEFINITIONS

Here we give a more complete and formal definition for models with tool-use. We start by defining a tool-use *oracle*, which will receive tool-use commands and will return an observation that corresponds to the execution of the command. This oracle will be stateful, meaning that its responses can vary depending on the *memory* of the oracle, which can be updated based on the commands it receives[11]. Let $\mathcal{M}$ be some set which will correspond to the set of memories of the oracle. We denote by $M_t \in \mathcal{M}$ the memory of the oracle after receiving $t$ commands, and let $M_0$ be the initial memory of the oracle. Importantly, we let the initial memory of the oracle depend on the input (e.g., the input can be stored in the memory of the oracle). For some memory $M_t \in \mathcal{M}$, we define $\mathcal{O}_{M_t} : \Sigma^* \to \Sigma^*$ to be the mapping from tool calls to observations, given memory $M_t$.

We augment the dictionary with additional tokens: $[\text{TOOL}], [\backslash\text{TOOL}], [\text{OBS}], [\backslash\text{OBS}],$ $[\text{THINK}], [\backslash\text{THINK}] \in \Sigma$. At any point in the generation, the model $h$ can issue a call to a tool by generating a sequence of the form $[\text{TOOL}], \boldsymbol{z}, [\backslash\text{TOOL}]$, for some $\boldsymbol{z} \in \Sigma^*$ which encodes the tool command. The command $\boldsymbol{z}$ is passed to the tool oracle, and the resulting observation that is then appended to the context of the model, with the format: $[\text{OBS}], \mathcal{O}_{M_t}(\boldsymbol{z}), [\backslash\text{OBS}]$. The model can also generate thoughts/reasoning, generated as a sequence $[\text{THINK}], \boldsymbol{z}, [\backslash\text{THINK}]$. All other tokens (tokens that are not tool-commands, observations or thinking tokens) are considered output tokens, and are appended to the output stream.

In the CoT-only setting, the model is only allowed to use thinking and output tokens. In the single-turn setting, the model can issue a single tool command, and can start generating the output after receiving the observation from the command (but can think before, after and during the output generation). In the interactive setting, a model can issue a tool call at any point in the generation, possibly interleaved with output tokens and tool commands. When evaluating the output, we ignore all tool commands, observations and thoughts and only consider the output stream at the end of generation.

# B PROOFS

## B.1 PROOF OF THEOREM 2.1

Before we introduce the complete proof of the theorem, we prove a simple version of the Theorem for the case where the output rule of the GSSM is deterministic. This proof is simpler than the more general stochastic setting, and captures the key principles for this result.

*Proof of Theorem 2.1 for deterministic GSSMs.* Let $\mathcal{S}$ be the state space of $h$. By definition, there exists some $n_0$ such that for all $n \geq n_0$ it holds that $\text{supp}_\alpha(f(\mathcal{D}_n)) > |\mathcal{S}|$. Now, fix some $n \geq n_0$, and let $A \subseteq \Sigma^*$ be all the possible outputs of $h$. Note that the output is determined by the state of $h$ before the output tokens are generated. Therefore, $|A| \leq |\mathcal{S}| < \text{supp}_\alpha(f(\mathcal{D}_n))$ and so we have

$$\Pr_{\boldsymbol{x}\sim\mathcal{D}_n}[f(\boldsymbol{x}) \in A] = \Pr_{\boldsymbol{y}\sim f(\mathcal{D}_n)}[\boldsymbol{y} \in A] = f(\mathcal{D}_n)(A) < \alpha$$

---

[11]For example, the oracle can receive a command for updating the content of a file on disk, which will affect its memory and hence future requests for reading the contents of the changed file.

Finally, we have

$$\Pr_{\boldsymbol{x}\sim\mathcal{D}_n}[f(\boldsymbol{x})=h(\boldsymbol{x})] = \Pr[f(\boldsymbol{x})=h(\boldsymbol{x})|f(\boldsymbol{x})\in A]\Pr[f(\boldsymbol{x})\in A]$$
$$+ \underbrace{\Pr[f(\boldsymbol{x})=h(\boldsymbol{x})|f(\boldsymbol{x})\notin A]}_{=0}\Pr[f(\boldsymbol{x})\notin A] \leq \mathcal{D}_n(A) < \alpha$$

and so $\operatorname{err}_n(h) \geq 1 - \alpha$. $\qquad\square$

We now show the proof of the theorem for the general (stochastic) setting:

*Proof of Theorem 2.1.* Let $\mathcal{S}$ be the state space of $h$, and we assume that $h$ operates either in the CoT-only or the single-turn setting. Denote by $U(\boldsymbol{x})$ the state of the model before generating the first output token. The model $h$ can generate thinking tokens and/or a single tool command before generating the first output token, and therefore $U(\boldsymbol{x})$ is a random variable that depends on $\boldsymbol{x}$ and the randomness of $h$.[12] Let $R(s)$ be the distribution of *outputs* (i.e. values of the output stream) generated by the model $h$ if it is at state $s$ before generating the first output token. Treating $h^{(\mathrm{out})}(\boldsymbol{x})$ as a random variable over outputs, we note that it depends only on the state after parsing $\boldsymbol{x}$, and therefore $h^{(\mathrm{out})}(\boldsymbol{x}) = R(U(\boldsymbol{x}))$. Additionally, we denote the conditional distribution over outputs induced by $h^{(\mathrm{out})}$ with $p(\boldsymbol{y}|\boldsymbol{x})$. Again, since the future generation of the model depends only on the state, we have $p(\boldsymbol{y}|\boldsymbol{x}) = p(\boldsymbol{y}|U(\boldsymbol{x}))$.

Now, by definition, there exists some $n_0$ such that for all $n \geq n_0$ it holds that $\operatorname{supp}_\alpha(f(\mathcal{D}_n)) > |\mathcal{S}|$. Fix some $n \geq n_0$. Fix some $s \in \mathcal{S}$. Let $\boldsymbol{y}_s$ be an output with maximal probability under the distribution $\mathcal{D}_n$, conditioned on the event that $U(\boldsymbol{x}) = s$:

$$\boldsymbol{y}_s = \arg\max_{\boldsymbol{y}} \Pr_{\boldsymbol{x}\sim\mathcal{D}_n}[f(\boldsymbol{x})=\boldsymbol{y}|U(\boldsymbol{x})=s]$$

Denote by $A$ the set of maximal probability outputs $A = \{\boldsymbol{y}_s \ : \ s \in \mathcal{S}\}$. Note that $|A| \leq |\mathcal{S}| < \operatorname{supp}_\alpha(f(\mathcal{D}_n))$ and so we have

$$\Pr_{\boldsymbol{x}\sim\mathcal{D}_n}[f(\boldsymbol{x})\in A] = \Pr_{\boldsymbol{y}\sim f(\mathcal{D}_n)}[\boldsymbol{y}\in A] = f(\mathcal{D}_n)(A) < \alpha$$

Observe that:

$$\Pr[f(\boldsymbol{x})=h(\boldsymbol{x})|U(\boldsymbol{x})=s] = \sum_{\boldsymbol{y}\in\mathcal{Y}_n} \mathbb{E}[1_{h(\boldsymbol{x})=\boldsymbol{y}} \cdot 1_{f(\boldsymbol{x})=\boldsymbol{y}}|U(\boldsymbol{x})=s]$$
$$= \sum_{\boldsymbol{y}\in\mathcal{Y}_n} \mathbb{E}[1_{R(U(\boldsymbol{x}))=\boldsymbol{y}} \cdot 1_{f(\boldsymbol{x})=\boldsymbol{y}}|U(\boldsymbol{x})=s]$$
$$= \sum_{\boldsymbol{y}\in\mathcal{Y}_n} \mathbb{E}[1_{R(s)=\boldsymbol{y}} \cdot 1_{f(\boldsymbol{x})=\boldsymbol{y}}|U(\boldsymbol{x})=s]$$
$$= \sum_{\boldsymbol{y}\in\mathcal{Y}_n} p(\boldsymbol{y}|s) \cdot \Pr_{\boldsymbol{x}\sim\mathcal{D}_n}[f(\boldsymbol{x})=\boldsymbol{y}|U(\boldsymbol{x})=s]$$
$$\leq \sum_{\boldsymbol{y}\in\mathcal{Y}_n} p(\boldsymbol{y}|s) \cdot \Pr[f(\boldsymbol{x})=\boldsymbol{y}_s|U(\boldsymbol{x})=s]$$
$$\leq \Pr[f(\boldsymbol{x})=\boldsymbol{y}_s|U(\boldsymbol{x})=s]$$

Where in the 4th equality we use the fact that the variables are independent. Therefore, we have:

$$\Pr_{\boldsymbol{x}\sim\mathcal{D}_n}[f(\boldsymbol{x})=h(\boldsymbol{x})] = \sum_{s\in\mathcal{S}} \Pr[f(\boldsymbol{x})=h(\boldsymbol{x})|U(\boldsymbol{x})=s]\Pr[U(\boldsymbol{x})=s]$$
$$\leq \sum_{s\in\mathcal{S}} \Pr[f(\boldsymbol{x})=\boldsymbol{y}_s|U(\boldsymbol{x})=s]\Pr[U(\boldsymbol{x})=s]$$
$$\leq \sum_{s\in\mathcal{S}} \Pr[f(\boldsymbol{x})\in A|U(\boldsymbol{x})=s]\Pr[U(\boldsymbol{x})=s] = \Pr[f(\boldsymbol{x})\in A] < \alpha$$

and so $\operatorname{err}_n(h) \geq 1 - \alpha$. $\qquad\square$

---

[12]We assume the oracle (e.g., the environment) is deterministic, but we can be easily extend this result to capture a stochastic oracle.

*Proof of Theorem 2.2.* First, let us define the oracle $\mathcal{O}$. The memory of the oracle at iteration $t$ holds a sequence of tokens $\boldsymbol{m}_t \in \Sigma^*$, and additionally some index $i_t \in \mathbb{N}$. At first iteration, we set $\boldsymbol{m}_0 = \boldsymbol{x}$ and $i_0 = 0$. The oracle $\mathcal{O}$ accepts the following commands:

- `read`: outputs the $i_t$-th token in $m_t$. If $i_t > |\boldsymbol{m}_t|$, output [EOS].

- `write` $\sigma$: updates the $i_t$-th token of $m_t$ to be $\sigma$.

- `move_left, move_right`: adds/subtracts 1 from $i_t$.

Next, we describe the training distributions $\mathcal{P}_n$. Since $f$ is tractable, there exists some Turing machine $\mathcal{T}$ that computes $f$. By definition, the machine halts for every input, and we can assume w.l.o.g. that it halts when the head is at position $0$. Let $Q$ be the (finite) set of states of $\mathcal{T}$, and let $q_0$ be the initial state. We assume the dictionary $\Sigma$ contains the following symbols: $\{0, 1, [\text{STATE}], [\backslash\text{STATE}]\} \in \Sigma$. For each state $q \in Q$, we define the encoding of the state $\text{enc}(q) = [\text{STATE}]\boldsymbol{z}_q[\backslash\text{STATE}]$, where $\boldsymbol{z}_q \in \{0, 1\}^{\log(|Q|)}$ is a binary encoding of the state. Then, for some input $\boldsymbol{x} \sim \mathcal{D}_n$, we construct a CoT of $\boldsymbol{x}$, denoted by $F(\boldsymbol{x})$, that will capture the "trace" of the machine $\mathcal{T}$:

- The sequence $F(\boldsymbol{x})$ begins with: $[\text{THINK}]\text{enc}(q_0)[\backslash\text{THINK}][\text{TOOL}]\texttt{read}[\backslash\text{TOOL}]$.

- In each step of the Turing machine processing $\boldsymbol{x}$, we add to $F(\boldsymbol{x})$ the sequence:

$$[\text{THINK}]\text{enc}(q)[\backslash\text{THINK}][\text{TOOL}]\texttt{read}[\backslash\text{TOOL}]$$
$$[\text{OBS}]\sigma[\backslash\text{OBS}][\text{TOOL}]\texttt{write}\,\sigma'[\backslash\text{TOOL}]$$

  where $q$ is the current state, and $\sigma'$ is the next symbol to write when reading $\sigma$ in state $q$. Additionally, we add $[\text{TOOL}]\texttt{move\_left}[\backslash\text{TOOL}]$ if the machine moves the head to the left, and otherwise $[\text{TOOL}]\texttt{move\_right}[\backslash\text{TOOL}]$.

- When the machine reaches a halting state, for every $i = 1 \ldots |f(\boldsymbol{x})|$ we add:

$$[\text{TOOL}]\texttt{move\_right}[\backslash\text{TOOL}][\text{TOOL}]\texttt{read}[\backslash\text{TOOL}][\text{OBS}]f(\boldsymbol{x})_i[\text{OBS}]f(\boldsymbol{x})_i$$

Note that since the machine computes $f(\boldsymbol{x})$ it will be written on its tape when it reaches a valid state. Therefore, it is easy to verify that the memory of the oracle $\mathcal{O}$ at step $t$ will hold the state of the tape and the correct position of the head, and that all the tool observations will be correct. Finally, $\boldsymbol{x} \sim \mathcal{D}_n$ and $F(\boldsymbol{x})$ together define the distribution $\mathcal{P}_n$ for all $n$, and it is indeed a training distribution for the task (since the non-tool tokens after the [ANS] token correspond to the correct output $f(\boldsymbol{x})$).

Next, we will show that a simple tool-SSM algorithm can achieve length generalization on this task. Let $\{(\boldsymbol{x}_1, F(\boldsymbol{x}_1)), \ldots, (\boldsymbol{x}_m, F(\boldsymbol{x}_m)\}$ be the set of examples observed by the algorithm. Let $\widehat{A}$ be the set of all pairs of state encodings and symbols that appear together in $F(\boldsymbol{x}_i)$ in some $\boldsymbol{x}_i$:

$$\widehat{A} := \{(q, \sigma) \,:\, \exists i,\, [\text{THINK}]\text{enc}(q)[\backslash\text{THINK}][\text{TOOL}]\texttt{read}[\backslash\text{TOOL}][\text{OBS}]\sigma[\backslash\text{OBS}] \in F(\boldsymbol{x}_i)\}$$

Note that for every $(q, \sigma) \in \widehat{A}$ there is a single symbol $\sigma'$ and a single command $\boldsymbol{d} \in \{\texttt{move\_left}, \texttt{move\_right}\}$ and a single state $q'$ that follow $(q, \sigma)$ in the trace (corresponding to the operation of the Turing machine). Let $R$ be the function mapping $(q, \sigma)$ to $(q', \sigma', \boldsymbol{d})$. Note that both $\widehat{A}$ and $R$ can be encoded with fixed (finite) memory. Therefore, we define a GSSM $h_{\widehat{A},R}$ that generates tokens as follows:

- Immediately after the input, generate: $[\text{THINK}]\text{enc}(q_0)[\backslash\text{THINK}][\text{TOOL}]\texttt{read}[\backslash\text{TOOL}]$.

- Following each response to a `read` command, generate:

$$[\text{THINK}]\text{enc}(q')[\backslash\text{THINK}][\text{TOOL}]\texttt{write}\,\sigma'[\backslash\text{TOOL}][\text{TOOL}]\boldsymbol{d}[\backslash\text{TOOL}]$$

  where $q', \sigma', \boldsymbol{d} = R(q, \sigma)$, for the $\sigma$ returned by the tool oracle.

- When a halting state is reached, generate the sequence:

$$[\text{TOOL}]\texttt{move\_right}[\text{\textbackslash TOOL}][\text{TOOL}]\texttt{read}[\text{\textbackslash TOOL}]$$

  and following the observation $[\text{OBS}]\sigma[\text{\textbackslash OBS}]$, output $\sigma$ (if $\sigma = [\text{EOS}]$ we halt the generation).

- If at some point we observe a pair $q, \sigma \notin \widehat{A}$, output $[\text{EOS}]$.

Denote by $A(\boldsymbol{x}) \subseteq Q \times \Sigma$ the set of state-symbol pairs observed by $\mathcal{T}$ when processing $\boldsymbol{x}$. Its easy to verify that for every $\boldsymbol{x}$ s.t. $A(\boldsymbol{x}) \in \widehat{A}$, the GSSM $h_{\widehat{A},R}$ will exactly recover $F(\boldsymbol{x})$. Therefore, the following lemma suffices for proving the theorem:

**Lemma B.1.** *Fix some* $\epsilon, \delta \in (0, 1)$. *There exists some* $n_0$ *s.t. with sample size* $m = n_0 |Q| |\Sigma| \log(|Q| |\Sigma| /\delta)/\epsilon$ *w.p. at least* $1 - \delta$ *over sampling from* $\mathcal{P}_1, \ldots, \mathcal{P}_{n_0}$ *it holds that:*

$$\forall n \geq n_0 \Pr_{\boldsymbol{x} \sim \mathcal{D}_n} \left[ A(\boldsymbol{x}) \subseteq \widehat{A} \right] > 1 - \epsilon$$

*Proof.* For every pair of symbols $\sigma \in \Sigma$ and state $q \in Q$, denote $p_n(\sigma, q) := \Pr_{\boldsymbol{x} \sim \mathcal{D}_n}[(q, \sigma) \in A(\boldsymbol{x})]$ the probability over sampling $\boldsymbol{x} \sim \mathcal{D}_n$ that the machine $\mathcal{T}$ reads a symbol $\sigma$ while it is in state $q$, when processing $\boldsymbol{x}$. Let $M := |Q| \cdot |\Sigma|$. Denote:

$$A_\epsilon = \left\{ (q, \sigma) \in Q \times \Sigma \text{ s.t. } \sup_n p_n(q, \sigma) \geq 2\epsilon/M \right\}$$

Now, for every $q, \sigma \in A_\epsilon$, let $n_0(q, \sigma)$ be the minimal $n$ s.t. $p_n(q, \sigma) \geq \epsilon/M$. Let $n_0 = \max\{n_0(q, \sigma)\}_{(q,\sigma) \in A_\epsilon}$. Let $m = \frac{n_0 M \log(M/\delta)}{\epsilon}$, and we will sample $m' = m/n_0 = \frac{M \log(M/\delta)}{\epsilon}$ examples from each of $\mathcal{D}_1, \ldots, \mathcal{D}_{n_0}$. Fix some $(q, \sigma) \in A_\epsilon$.

**Claim:** w.p. at least $1 - \delta/M$ we have $(q, \sigma) \in \widehat{A}$.

**Proof:** Note that $n_0(q, \sigma) \leq n_0$, and therefore we sample $m'$ examples from $\mathcal{D}_{n_0(q,\sigma)}$. Let $p := \Pr_{\boldsymbol{x} \sim \mathcal{D}_{n_0(q,\sigma)}}[(q, \sigma) \in A(\boldsymbol{x})]$ and by definition $p \geq \epsilon/M$. Therefore, for the $m'$ samples we draw, the probability that we do not encounter $(q, \sigma)$ in any of the traces is at most $(1 - p)^{m'} \leq (1 - \epsilon/M)^{m'} \leq \exp(-m'\epsilon/M) \leq \delta/M$.

From the above claim, using the union bound, we get that w.p. at least $1 - \delta$ we have $A_\epsilon \subseteq \widehat{A}$. Assume this holds, and fix some $n \geq n_0$. For every $(q, \sigma) \in Q \times \Sigma \setminus A_\epsilon$ it holds that $\Pr_{\boldsymbol{x} \sim \mathcal{D}_n}[(q, \sigma) \in A(\boldsymbol{x})] \leq \epsilon/M$. From the union bound, the probability over $\boldsymbol{x} \sim \mathcal{D}_n$ that there exists some $(q, \sigma) \notin A_\epsilon \subseteq \widehat{A}$ s.t. $(q, \epsilon) \in A(\boldsymbol{x})$ is at most $|Q \times \Sigma \setminus A_\epsilon| \epsilon/M \leq \epsilon$. Therefore, the required follows. □

From the above lemma the proof of Theorem 2.2 follows.

□

## C ARCHITECTURE AND TRAINING DETAILS

We train the following architectures for the synthetic experiments:

- **Mamba-130M** (https://huggingface.co/state-spaces/mamba-130m-hf): a selective state-space (SSM) language model. We use a 24-layer, 1536-d intermediate size and 768-d model size configuration to match the Transformer baselines while retaining linear-time sequence modeling.

- **LSTM**: a multi-layer recurrent baseline sized to roughly comparable capacity (4 layers, hidden size 1536) to probe how classical RNNs fare on our trajectory-style tasks.

- **GRU**: a gated-recurrent baseline (4 layers, hidden size 1536) offering a stronger RNN comparator with fewer parameters per unit than LSTM.

- **Pythia (GPT-NeoX style)** (`https://huggingface.co/EleutherAI/pythia-160m`): a decoder-only Transformer from the Pythia scaling suite. We adopt a 24-layer, 1536-d intermediate size, 768-d model size and 8-head model variant with RoPE, roughly matching Mamba's scale.

- **Mistral-style Transformer** (`https://huggingface.co/mistralai/Mistral-7B-v0.1`): a modern decoder-only Transformer with sliding-window (512) sparse attention, utilizing RoPE. We use a scaled-down 8-layer, 1536-d intermediate size and 768-d model size.

For the synthetic experiment, we perform hyper-parameter search over learning rate, batch size and weight decay. We choose $\text{learning\_rate} \in \{0.00005, 0.0001, 0.0003, 0.0005, 0.001, 0.003, 0.005\}$, $\text{batch\_size} \in \{128, 256, 512, 1024\}$, $\text{weight\_decay} \in \{0, 0.01\}$ and fix the number training steps to be $2,000$. We run each experiment with 2 seeds, and report the accuracy of the best model. For Tower of Hanoi experiments, due to their sensitivity, we exceptionally used 10 seeds for Mamba and Pythia models and 3 seeds for other architectures. For the code fixing experiment, we finetune a pretrained Mamba-1.4b and Pythia-1.4b, both trained on The Pile (Gao et al., 2020), with learning rate 0.0001, weight decay 0.01, batch size 512 and 200 training steps. For all experiments, we use a single node with 8 H100 GPUs.

## D  SYNTHETIC EXPERIMENTS DETAILS

### D.1  MEMORY TOOL DEFINITIONS

As discussed in Section 3, we use either a pointer-based or a search-based memory tool to augment the memory of the model. We now describe how the model interacts with the memory tool, and how we differentiate between *thoughts*, *outputs*, *commands* and *observations*. We generally try to reduce the number of tokens by using dedicated command tokens, and differentiate between output, thoughts and observation tokens based on the context (instead of using open/closing tags).

**Pointer-based Memory.**  The commands for this memory are given as special tokens that the model can output, e.g. [pointer1.read()] or [pointer2.move_left()]. A read command will be immediately followed by a single *observation* token, that is the token read by the pointer at its current position. All other tokens (tokens that are not command tokens or observation tokens, which always immediately follow a read command) are either *thoughts* or *outputs*. We use a single token [ANS] that indicates the final answer of the model, where all tokens before the [ANS] token are considered *thoughts* and all tokens after the [ANS] token are considered *outputs*. Both *thoughts* and *outputs* are appended to the context memory, and the pointers can move beyond the input context, and start reading *thought* or *output* tokens that were previously generated by the model. *Commands* and *observations* are discarded and are not appended to the external memory (but of course do affect the internal memory and representation of the model). The model can freely interleave commands, thoughts and outputs, and therefore the model can interact with the memory while producing the answers.

**Search-based Memory.**  This memory tool allows the model to search for a given pattern in the context. A search command is a sequence of tokens of the form: [COMMAND]find[VALUE]$x$, where $x$ is some sequence of tokens to search for. Following the search command, the model will receive a set of *observations* of the form: [OBSERVATION]$z_1, \ldots, z_k$, where $z_1, \ldots, z_k$ are all the lines in the memory context that contain the string $x$ (similar to the operation of a `grep` command). As before, all other tokens are either *thoughts* or *outputs*, and are appended to the memory and can be searched for in future iterations. In this case we take the output to be the last line generated by the model.

### D.2  LOGICAL REASONING TASK

As described in Section 3, we generate a random logical computation graph with $k = 3$ input nodes, where each intermediate node is a boolean expression over one or two variables or their negation. The graph is encoded as a python code given to the model as input. Illustration of the graph and the code are shown in Figure 4.

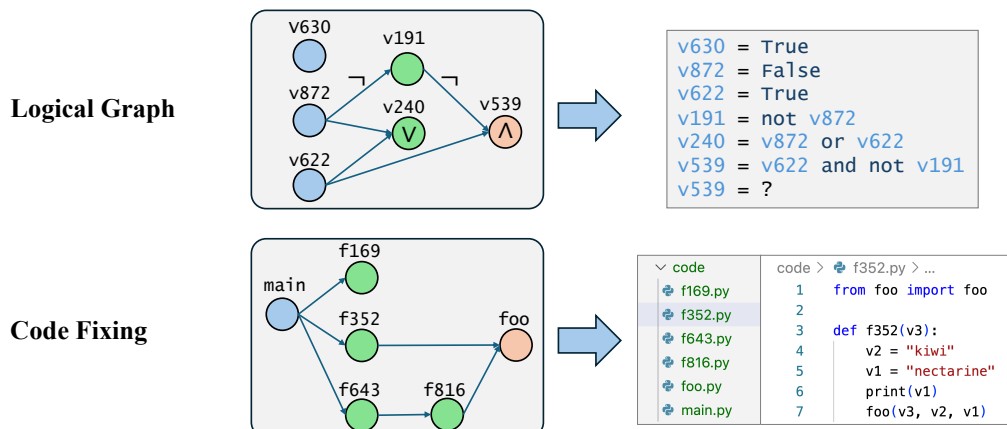

Figure 4: Illustration of Logical Graph reasoning task and code fixing task. We generate random graphs that define logical function structure (top) or code dependencies (bottom), and synthetically generate problems according to these graph structures.

### D.3 Tool-Use Algorithms

We describe the synthetically generated tool-use trajectories for solving the different tasks presented in Section 3.

**Multi-Digit Addition.** We follow the standard long addition algorithm, summing one-digit from right to left while keeping the "carry" digit. The model uses pointer-based memory with two pointers, and performs the following steps:

1. Move each pointer to the least significant digit of each summand, where the first pointer points to the digit of the first summand, and the second pointer to the second summand. To do this, we move the first pointer until we read the token $+$, and move the second pointer until we read the token $=$.

2. Read one digit from each summand, compute the sum of the digits and the carry from previous iteration (if it exists), output the new sum and carry as *thoughts*, and move both pointers to the left. Stop each pointer if it reaches a non-digit token. If both pointers reached non-digit tokens, output [ANS] and move to the next step.

3. At this step we have the sum written in reverse in memory, along with the carry digits from each iteration. To write the final output in reverse, we move the second pointer to the right until it reaches the [ANS] token, then start moving to the left, each iteration outputting the sum digit, until the pointer reaches the $=$ token.

**Multi-Digit Multiplication.** We follow the long multiplication algorithm for multiplying an $n$-digit number by a $k$-digit number (for fixed $k$). We use a pointer-based memory with $\max(k, 2)$ pointers. The algorithm executes the following steps:

1. Move the first pointer to the least significant digit of the first operand, and the second pointer to the least significant digit of the second operand.

2. Move the first pointer to the left, each time reading a digit and multiplying it with the digit of the second pointer. Add the result to the previous carry, and write it together with the new carry. If we reach the most significant digit of the first operand, move the first pointer back to the least significant digit, move the second pointer one position to the left, output a $+$ sign and zeros as required (depending on which digit from the second operand we read). If the second pointer reached the $\times$ sign, move to the next step.

3. At this step we have a summation problem with $k$ summands, where the summands are written in reverse and also contain carry digits that we should ignore. We move each pointer to the least significant digit of its respective summand, read all digits, compute the

      sum and the carry, and move each pointer to the right and skip carry digits. We continue until we reach the most significant digit of all summands, and then output an [ANS] token.

4. Finally, we have the answer written in reverse with carry digits. We move the first pointer to the [ANS] token, then move it one token to the left and output the tokens read by the pointer, skipping carry digits.

**Tower of Hanoi.** The Tower of Hanoi puzzle can be solved by a simple recursive algorithm. Let $n$ be the number of disks in the puzzle. The recursive algorithm involves three steps: (1) recursively moving the top $n-1$ disks from rod $A$ to rod $B$; (2) moving the largest disk from rod $A$ to rod $C$; and (3) recursively moving the $n-1$ disks from rod $B$ to rod $C$. Therefore, the puzzle can be solved with $2^n - 1$ moves. This algorithm can also be stated iteratively:

- At the first step, the smallest disk is moved from rod $A$ to rod $B$ or $C$ depending on whether $n$ is even or odd, respectively.

- At the second and other even steps, the only legal move not involving the smallest disk is performed.

- At the third and other odd steps, the smallest disk is moved to the rod it was not on two turns ago.

Our model uses the iterative algorithm described above to solve the puzzle. The model uses pointer-based memory with one pointer. At each step of the algorithm, the model outputs the next move and the subsequent state of the rods. Specifically, the model takes the following steps:

1. The pointer traverses rod $A$ from its base, reading disks one by one and outputting $B$ and $C$ alternatively. This step computes the parity of $n$, which is crucial for the first move. After reaching the end of rod $A$, the pointer rewinds to the beginning of the current state representation. The model is now ready to predict the next move.

2. At this step, the model outputs the next move, e.g., $(5)AC\$$[13] (meaning disk (5) is moved from rod $A$ to rod $C$). Next, the model outputs the new state of the puzzle and goes to the step described below. In case that all disks are on rod $C$, no move is predicted and the output ends at this step.

3. The model uses the pointer to output the new state. The pointer starts by reading the previous state one disk at a time, outputting the new state. Note that the new state differs from the previous state only by the position of a single disk. After processing the previous state, the pointer is then advanced past the outputted move (e.g., (5)AC\$) to position itself at the beginning of the newly generated state. The model then goes back to the move prediction step above.

We also trained our models using the recursive algorithm; however, their length generalization performance was weaker. Details of this experiment are presented in Appendix D.4.

**Logical Reasoning.** We use a search-based memory tool to solve the logical reasoning problem detailed in Appendix D.2. We try to resolve variables' truth value recursively using depth-first-search (DFS). Namely, starting with the output variable, we recursively search for the values of variables in a given expression. If we find a variable with a boolean (True/False) value, we update the expression, replacing the variable's name by its value. If we find a child variable that is still not resolved, we search for the variables in the child's expression, while also logging the value of the parent variable (that we can use for "backtracking"). When we are done resolving a variable's value, we backtrack to its parent, trying to resolve the parent's value. When we resolved the output nodes value, we finish the generation.

## D.4 TOWER OF HANOI EXPERIMENT DETAILS

In contrast to other tasks presented in this paper, the solution length for the Tower of Hanoi puzzle scales exponentially with the number of disks. In particular, solving the puzzle for 9 and 12 disks

---

[13]$ is used after each move as a delimiter.

Table 2: Experimental results for the Tower of Hanoi puzzle solved recursively by different models.

| Model | Hanoi (recursive) |
|---|---|
| Mamba | 7→8 (100%) |
| LSTM | **7→9 (83%)** |
| GRU | 7→8 (100%) |
| Pythia | 7→7 (100%) |
| Mistral | 7→8 (87%) |

would require over 42,000 and 385,000 tokens, respectively. A solution is considered correct only if all of its tokens are correctly generated. As a result, even if a model has 99% token accuracy, it may not show any length generalization capability. In agreement with this intuition, we found that Hanoi length generalization performance is very sensitive to the random seed. Hence, we used 10 seeds for Mamba and Pythia models and 3 seeds for the rest of the models. In our experiments with Mamba models, we noticed that token accuracy is always high (e.g., $\geq 99.75\%$ for 12 disks), however, the actual accuracy varies based on the seed. The performance of our 10 seeds (all trained on puzzles with up to 8 disks) is shown in Figure 5. We note that Pythia did not show any length generalization, in particular, even the token accuracy did not exceed 93%.

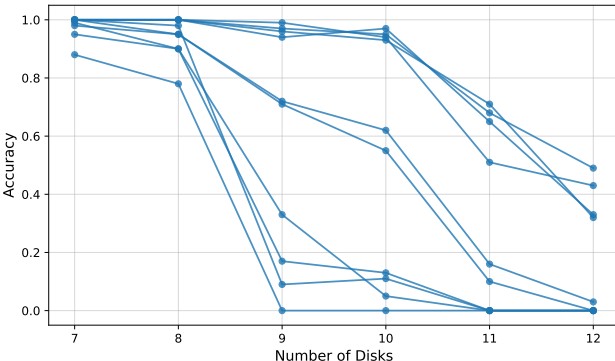

Figure 5: Performance of different seeds when training a Mamba model on Tower of Hanoi puzzles with up to 8 disks.

As discussed in Appendix D.3, the Tower of Hanoi puzzle can also be solved recursively. We also tried the recursive variant of the algorithm, however, its length generalize performance was weaker than the iterative algorithm. Here, we include the implementation and results of the recursive implementation.

**Recursive Implementation.** For a puzzle with $n$ disks, the model outputs the list of moves for puzzles of size 1, 2, …, $n$ sequentially and uses the move list generated for puzzle of size $i - 1$ to output the list of moves for puzzle of size $i$ using the recursive pattern. The model uses pointer-based memory with two pointers. While outputting the list of moves for the puzzle of size $i$, the first pointer points to the $i$th disk (largest disk moved in the moves of the puzzle of size $i$). The second pointer is used for implementing the recursive pattern and iterating the moves of the puzzle of size $i - 1$ while generating the moves for size $i$. More precisely, the input gives the list of disks (e.g., $(7)(5)(2)$) and the model executes the following steps:

1. Both pointers are moved to the smallest (top) disk, and the model outputs the first move, i.e., moving the top disk from rod $A$ to rod $C$. This solves the problem for a single disk. First pointer moves one step back (now pointing to the second smallest disk) and the second pointer advances, pointing to the beginning of the first move. The model is now ready to output the moves for solving the puzzle for two disks.

2. At this step, the model copies the last list of moves, swapping rod labels $B$ and $C$. To achieve the latter, the second pointer traverses the last list of moves and the model reads

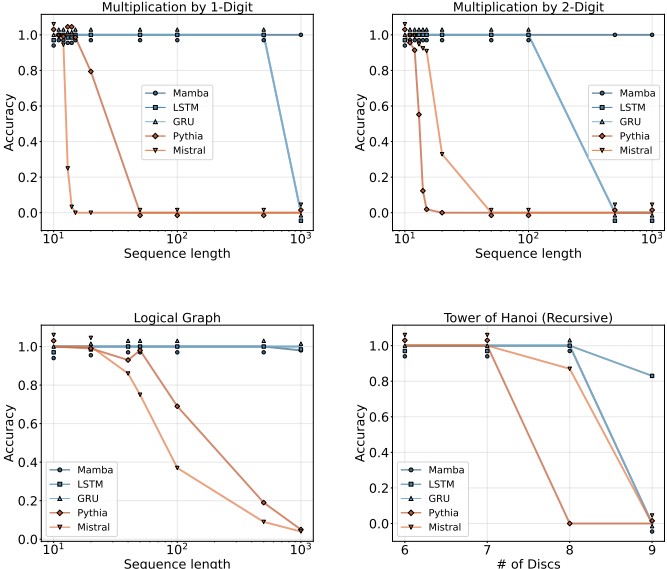

Figure 6: We train various transformer (Pythia, Mistral) and SSM (Mamba, LSTM, GRU) models on Multi-Digit Multiplication, Logical Graph and Tower of Hanoi tasks, with CoT + pointer-based memory tool. **Multi-Digit Multiplication:** We train models on multiplying a number of up to 10-digit by a 1-digit number or 2-digit number, using the pointer-based memory tool. **Logical Graph:** We train models to perform a logical graph reasoning problem using search-based memory tool, training on graphs with up to 10 variables. **Tower of Hanoi:** We train models to solve the *Tower of Hanoi* (recursive implementation) reasoning problem using search-based memory tool, training on problems with up to 7 disks. The first point in each plot is the maximal problem size seen during training (i.e., all other points are out-of-distribution extrapolation).

and outputs one token at a time (performing the swap if needed). This step corresponds to the first step of the recursive algorithm. At the end of copying, the second pointer is rewound to point to the beginning of the list of moves again.

3. Next, the middle move, i.e., moving the largest disk ($i$th disk while outputting the moves of size $i$) is performed. This disk is identified by the value of the first pointer and the move is always from rod $A$ to $C$. This step corresponds to the second step of the recursive algorithm.

4. Similar to step 2, the model copies the list of moves again, swapping $B$ and $A$. The second pointer is used again for iterating the list of moves and copying. This step corresponds to the third step of the recursive algorithm. After the copying is finished, the second pointer advances and points to the beginning of the newly constructed move list. The first pointer goes one step back so that it points to the next larger disk. This completes the generation for size $i$, and the process iterates by returning to step 2 for size $i+1$. The generation terminates if there is no disk remaining for the first pointer, indicating that the lists of moves have been generated for all puzzle sizes $1, \ldots, n$.[14]

The performance of the recursive solution for the Tower of Hanoi puzzle is reported in Table 2.

### D.5 SSMs and Transformer Baselines

In Figure 6 we report accuracies of our baseline models on the Multi-Digit Multiplication, logical reasoning and Tower of Hanoi tasks. We train each model on the same trajectories, using CoT and tool use. We perform hyperparameter optimization for each model as described in C. Our results

---

[14]We note that we use a delimiter (e.g., #) between the list of moves for different number of disks so that they become separable.

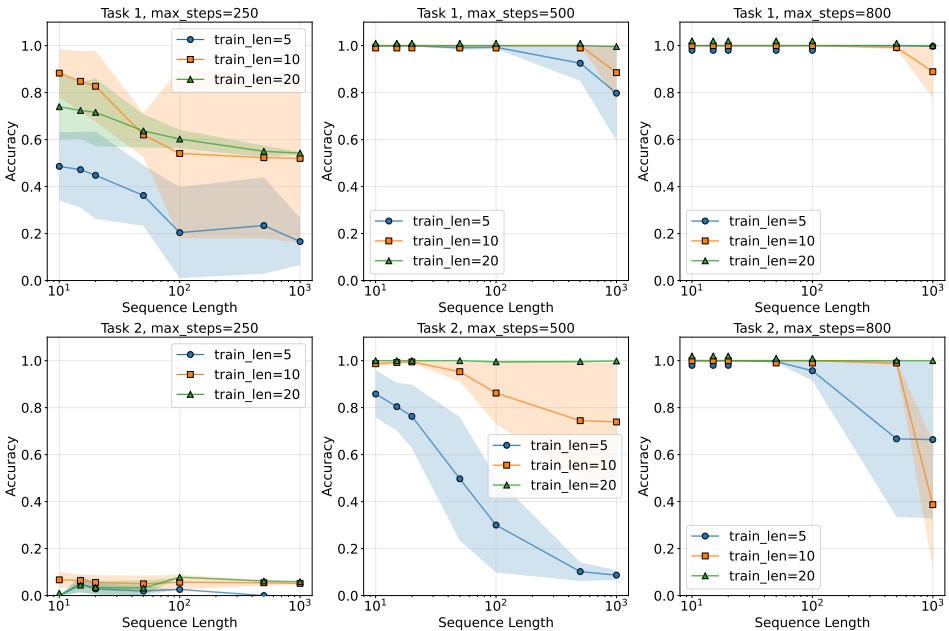

Figure 7: Multiplication generalization performance for Mamba across different training configurations. Each subplot shows accuracy as a function of sequence length for a specific maximum training steps value. Different colored lines represent different training sequence lengths, with error envelope indicating median absolute discrepancy across 5 runs.

generally point to a length generalization advantage for state space models over baseline transformer models.

### D.6 ABLATING TRAINING STEPS AND DIGIT LENGTH FOR MULTIPLICATION

In Figure 7 we investigate the impact of different training configurations on generalization for multi-digit multiplication, varying the training budget (250, 500, or 800 steps) and the maximum number of digits seen during training for the first operand (5, 10, or 20). For this experiment, the learning rate was set to 0.003 based on validation. Results indicate that increasing the maximum number of digits shown during training for the first operand improved stablity of OOD generalization consistently, with results improving with more training steps. In particular, training with up to 20 digits improves generalization stability perfectly up to the maximum digit size tested (1000 digits). However, even training with up to 5 and 10 digits show progressive improvements as the number of training steps increases.

### D.7 ADDITIONAL ABLATIONS

We run the following ablations on the multi-digit addition task:

1. No-CoT: we train the model to directly output the final answer, without any CoT or tool-use.

2. No-CoT, reversed answer: we train the model to directly output the final answer in reverse (reverse format was shown to improve length generalization in certain settings, e.g. Zhou et al. (2024)).

3. No Tool-Use: The model is trained on similar trajectories as in the main experiment, but now needs to predict the output of the memory tool instead of receiving these as observations. Namely, the trajectory is used as CoT data.

4. Single-Turn Tool-Use: we train the model with a "calculator", where the model needs to generate a single addition command following the input (i.e., given an input $a + b$ it needs to generate $\text{add}(a, b)$).

We train the Mamba model in all settings with extensive hyper-parameter tuning on 5-digit addition. Experiments 1,2 and 4 results in perfect accuracy on 5-digit addition, but little to no length generalization. Experiment 3 results in poor performance even in-distribution.

## D.8  TASK MIXTURE

In Figure 8 each panel shows accuracy as a function of test length for various training budgets (250, 500, or 800 steps). The curves correspond to different mixing weights, where $w = 0$ denotes the baseline trained only on the main task and higher values indicate a normalized fraction of auxiliary samples. The error bars indicate variability across random seeds.

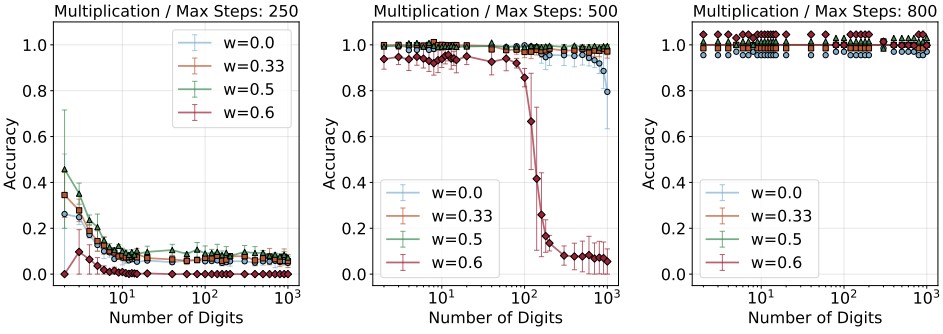

Figure 8: Multiplication task accuracy under co-training with varying training budgets (see Sec 3.1).

The accuracy plots for the main task (multiplication) in the task mixture experiment were presented in section 3.1. For completeness, we show the auxiliary task accuracy in Figure 9.

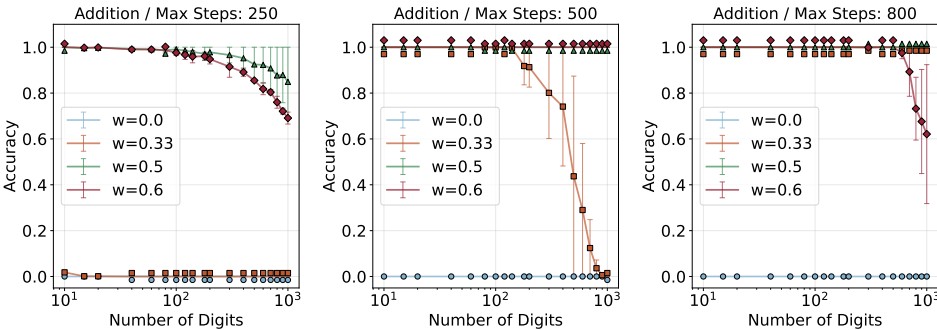

Figure 9: Addition task accuracy under co-training with varying training budgets (250, 500, 800 steps). Curves show different mixing weights. See (3.1).

## D.9  COMPARISON WITH HYBRID-MAMBA AND RMT

In addition to comparing Mamba against Transformers, we also evaluate two alternative architectures on the multi-digit addition task with CoT and pointer-based memory tool: Hybrid-Mamba and the Recurrent Memory Transformer (RMT) (Bulatov et al., 2022).

**Hybrid-Mamba.**  The Hybrid-Mamba model follows the Jamba architecture Lieber et al. (2024), which interleaves Mamba SSM layers with periodic Transformer attention layers. In our configuration, we use a 24-layer model with hidden size 768 and intermediate size 2048, where every 4th layer is a standard Transformer attention layer (with 12 attention heads and 4 key-value heads) and the remaining 20 layers are Mamba SSM layers (with state size 16, convolution kernel size 4, and expansion factor 2).

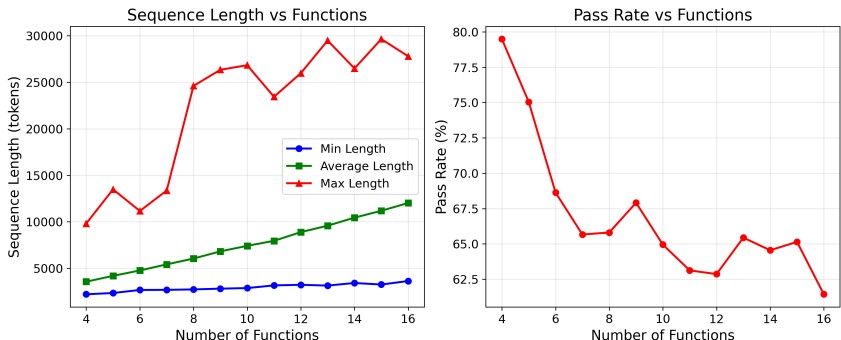

Figure 10: Pass rate and median sequence length for SWE-agent-LM-32B on the code fixing task.

**Recurrent Memory Transformer (RMT).**   The RMT model (Bulatov et al., 2022) augments a GPT-NeoX Transformer (24 layers, hidden size 768, 8 attention heads, intermediate size 1536) with learnable memory tokens and segment-level recurrence. The input is divided into segments of fixed length, and each segment is processed with the layout: [read_mem | content | write_mem]. The read-memory tokens are prepended to each segment and attend bidirectionally among themselves; content tokens attend causally to themselves and fully to the read-memory tokens; write-memory tokens attend fully to both read-memory and content tokens, and causally among themselves. The write-memory hidden states from segment $k$ become the read-memory inputs for segment $k + 1$, enabling information to flow across segments through a fixed-size memory bottleneck. We use 10 memory tokens and experiment with a segment length of 16 and 64.

**Experimental Setup.**   We train all three model types (Mamba, Hybrid-Mamba, and RMT) on the addition task with CoT and pointer-based memory tool, training on up to 5-digit addition and evaluating on longer sequences.   We perform a hyperparameter sweep over learning rate $\in \{0.00005, 0.0001, 0.0003, 0.0005, 0.001, 0.003, 0.005\}$, effective batch size $\in \{128, 256, 512, 1024\}$, and weight decay $\in \{0, 0.01\}$, with a fixed budget of 2,000 training steps.

**Results.**   We find that the Hybrid-Mamba model performs similarly to Mamba, achieving perfect length generalization on the addition task, extrapolating from 5-digit to 1,000 digit addition. This suggests that the periodic attention layers do not hurt the length generalization capability of the SSM backbone. In contrast, while the RMT model learns the task in-distribution (achieving 100% accuracy on 5-digit addition), it does not achieve meaningful length generalization.

## E   CODE FIXING AGENT SETUP

We use the same system prompt and input prompt as in mini-SWE-agent (Yang et al., 2024a) from: `https://github.com/SWE-agent/mini-swe-agent`. We instruct the model to solve the bug in `main.py`, and explain how the bug should be solved. We modify the original prompt of mini-SWE-agent to instruct the model interactively debug the code and generate a fix for up to 3 files at a time.

> *Please solve this issue: Fix the bug in main.py. Make sure to pass variable v10 too foo() and all other relevant functions. Pass v10 to ONLY the relevant functions, do not pass it if it is not needed.*
>
> *You can execute bash commands and edit files to implement the necessary changes.*
>
> *## Recommended Workflow*
>
> *This workflows should be done step-by-step so that you can iterate on your changes and any possible problems.*
>
> *1. Create a script to reproduce the issue and run it*
>
> *2. Spot 3 files that might be causing the issue*

*3. Read the content of these 3 files.*

*3. Edit the source code of these files resolve the issue. Do not edit more than 3 files before running the script again, even if the code is not completely fixed.*

*4. Verify your fix works by running your script again, if not - analyze at most 3 more files that might cause the issue and repeat the debugging process*

*5. Submit your changes and finish your work by issuing the following command: 'echo COMPLETE_TASK_AND_SUBMIT_FINAL_OUTPUT'. Do not combine it with any other command. ⟨important⟩After this command, you cannot continue working on this task.⟨/important⟩*

We plot the pass rate and generated trajectory length of the SWE agent as a function of the number of functions in the code in Figure 10.

## F    TOOL USE CAPABILITIES OF PRETRAINED SSMS

At the time of writing, we were unable to find any publicly-available SSM models that were fine-tuned for function calling. The closest we could find is Mistral's Mamba-Codestral-7B-v0.1, which was fine-tuned on coding tasks. We evaluated this model on the Berkeley Function Calling Leaderboard (Patil et al.), and found an overall accuracy of 16.58%, comparable with the reported accuracies of 16.22% for Falcon3-3B-Instruct and 15.58% for Llama-3.1-8B-Instruct.

