# OpenReview forum: "To Infinity and Beyond: Tool-Use Unlocks Length Generalization in State Space Models"
_ICLR.cc/2026/Conference — ICLR 2026 Oral_

### Official Review · Reviewer_Mthj · 2025-10-17

**Soundness:** 3
**Presentation:** 4
**Contribution:** 4
**Rating:** 8
**Confidence:** 5

**Summary:**

This paper studies **length generalization** of **State Space Models (SSMs)** and recurrent controllers on long-form generation tasks. It formalizes a class of **generalized SSMs (GSSMs)**, that are a finite-state controllers whose memory does **not** grow with sequence length, and a family of **long-form generation** problems where the support size of valid outputs increases with task complexity. Two central theoretical results:

* **Impossibility (Thm 2.1):** Any CoT-only or **single-turn** tool-use GSSM must fail on long-form generation beyond some size with a specific error.
* **Possibility (Thm 2.2):** With **interactive** tool-use (a simple read/write pointer-tape oracle), there exist training distributions under which a GSSM **length-generalizes** on any *computationally tractable* long-form task, i.e., learns a constant-state controller that operates unbounded external memory.

Empirically, the paper shows that SSMs/RNNs trained on **interactive, ReAct-style** trajectories strongly extrapolate on:

* synthetic arithmetic (multi-digit addition/multiplication up to 1K digits),
* reasoning (logical DAG evaluation to 1K nodes; some generalization on Tower of Hanoi),
* a **coding** bug-fix task with bash/file tools (run→inspect→patch→re-run), where Mamba trained on **interactive/distilled** traces maintains high pass rates well beyond training repository sizes, while **single-turn** tool-use collapses and local-attention Transformers degrade.

**Contributions:**

1. A clean **formalization** of long-form generation and GSSMs.
2. A **negative result** showing fixed-memory controllers (incl. CoT or single-turn tool calls) cannot solve such tasks at scale.
3. A **positive result**: interactive tool-use + external memory is sufficient for “infinite” length generalization.
4. Broad **experimental evidence** across arithmetic, reasoning, and coding, with a realistic agentic setup and trajectory distillation.

**Strengths:**

* **Originality:** Elegant formalization of long-form generation via support-growth; clean GSSM abstraction; sharp **necessary** (no interaction) vs **sufficient** (interactive tools) results.
* **Quality:** Both **theory and practice** reinforce each other; experiments span arithmetic, graph reasoning, and a realistic coding benchmark with real tool commands and agent distillation.
* **Clarity:** Clear separation of CoT-only, single-turn, and interactive regimes; easy-to-follow trajectory visualizations.
* **Significance:** Offers a compelling recipe for **length-generalizing agents**: small SSM controller + interactive memory/tooling + trajectory imitation. Practical implications for efficient, long-horizon tasks.

**Weaknesses:**

* **Baselines & fairness:** Transformer comparisons are mainly **local/sliding-window**. Missing head-to-head with **memory-augmented** or **recurrent** Transformer variants (e.g., retrieval-augmented, ring/segment memory, recurrent attention, kNN-LM, RMT) that also enable externalized state.
* **Policy-execution gap:** Training uses teacher forcing on trajectories; it would help to quantify compounding-error effects when **executing** the learned policy end-to-end under stochastic observations.
* **Scaling breadth:** Coding experiments use 1.4B-scale models; it would be informative to include **larger SSMs** and show whether gains persist or amplify vs similarly scaled Transformer agents with function-calling tool APIs.

**Questions:**

1. **Scope of Theorem 2.2:** The result asserts existence of training distributions {P_n}. How constructive is the procedure? Can you characterize **sample complexity** (number of transitions/examples) needed to learn a given algorithm’s controller?
2. **Transformers + external memory:** How do results change when comparing against **Transformers with the same pointer-tape tool** and **interactive** protocol? Are failures due to representation (attention) or optimization/data?
3. **Tool noise & latency:** Real tools return noisy, delayed, or partially structured outputs. How sensitive is the controller to **observation noise**, **command failures**, or **latency**?
4. **Generalization beyond seen transitions:** For multiplication/addition, do models truly learn **state transition rules**, or are there subtle failure modes (e.g., rare carry chains)? Any adversarial or **worst-case** tests?
5. **Safety & security:** In coding, could the learned policy execute unsafe commands? Any recommended **sandboxing** or **guardrails** for deploying SSM agents with system tools?
6. **When is single-turn enough?** Can you give some task classes where single-turn (or CoT-only) suffices, perhaps when supp_alpha grows slowly or when a single observation is maximally informative?

**Details Of Ethics Concerns:**

No concerns.

---

> ### Author Response · Authors · 2025-11-20
>
> We thank the reviewer for the very positive feedback on our paper, and for their helpful comments. Below is a response to the different weaknesses and questions raised by the reviewer:
>
> Weaknesses:
>
> * Other baselines: we compare SSMs to both standard full-attention Transformer (the Pythia architecture) and a sliding-window attention (Mistral architecture). We are happy to compare to other baselines that combine attention and external memory. However, there are many works that propose different variants of attention that includes externalized state, without clear winner, and it would be difficult to compare to all these alternatives. If there is a specific architecture that you believe would be a good candidate for comparison, please share your suggestion.
> * Policy-execution gap: In our synthetic experiments, we measure the correctness of all steps in the process, so a correct example means that there were no errors during the execution. In the coding experiment, we measure the correctness of the fixed code, and the model can potentially recover from errors during the execution. In all cases, the environment is deterministic, so there is not stochasticity in the observations.
> * Larger-scale models: When training models from scratch, we limited experiments to relatively small models so that they can be easily reproduced without significant computational burden. We also experimented with pretrained 1.4B SSM and Transformer models, which show trends that are similar to what we observe for small models. We believe that we will see similar behavior in larger scale models. Indeed, different works [1,2] have shown that even frontier transformer-based models struggle with tasks that involve long-context or long-form generation. Unfortunately, there are currently no frontier-level SSM-based model that we can compare these models against, and training one is beyond the scope of this work.
>
> Questions:
>
> 1. Theorem 2.2: The construction of the distribution depends on the Turing machine, so if we know the Turing machine for performing the computation, we can construct the distribution (as shown in the proof of the theorem). Regarding sample complexity: in the proof of Lemma B.1 we derive the sample complexity bound, which depends on the value of the minimal task complexity n_0 as well as the size of the Turing machine. We will add this sample complexity to the statement of the result.
> 2. Transformers + external tools: in all our experiments, we train all models in the same setting, so Transformers are given the same tool access as SSMs. We believe that the failure of Transformers is due to issues in out-of-distribution generalization, perhaps due to a wrong inductive bias and performance degradation due to long context.
> 3. Tool noise and latency: we did not experiment with noisy observation, and believe that this is a fascinating topic for future work. As for latency in tool-use, this will mainly affect the runtime of the model, potentially making inference slower and therefore diminishing the benefit of using SSMs over Transformers. We will therefore add a discussion on this point, acknowledging that the benefit of SSMs with tool-use depends on the exact setting, environment and solution engineering.
> 4. Generalization beyond seen transition: In our multiplication and addition experiments, we see that SSMs extrapolate remarkably well, from training on 5 or 10-digit operations to 1000-digit operations. This demonstrates that the models learn the true transition rule in the vast majority of cases. We cannot rule out the possibility that there are worst-case examples, but focus in this work on average-case performance.
> 5. Safety and Security: as in any setting that involves coding agents, models can potentially execute unsafe commands. There are various solutions and safety measures for handling this, which are beyond the scope of this paper, as we do not believe that SSMs would pose a different risk compared to standard models.
> 6. Single-turn or CoT-only can be sufficient for SSMs if the size of the output space is fixed as a function of the input length (i.e., tasks that do not involve long-form generation), and the required memory for solving the task is fixed. There are many tasks of this form, for example arithmetic tasks like computing a sum of numbers modulo p or binary classification tasks that require fixed memory.
>
> [1] Shojaee et al., 2025. The Illusion of Thinking: Understanding the Strengths and Limitations of Reasoning Models
> via the Lens of Problem Complexity.
> [2] Bertsch et al., 2025. Oolong: Evaluating Long Context Reasoning and Aggregation Capabilities

---

> ### Comment · Reviewer_Mthj · 2025-11-24
> **Additional experiments**
>
> For a concrete baseline, I’d be especially interested in a comparison to a memory-augmented Transformer, such as the Recurrent Memory Transformer (RMT; Bulatov et al., NeurIPS 2022), which incorporates recurrent memory tokens and is explicitly designed for long-context and algorithmic tasks. A smaller-scale RMT backbone with the same interactive tool protocol would be a very direct test of whether your gains come from the SSM inductive bias versus “just” giving a Transformer externalized state. Additionally, models such as Memformer [1] or Associative Recurrent Memory Transformer (ARMT) [2] would also fit your “controller + external memory” framing while keeping the set of baselines manageable.
>
> [1] Memformer: A Memory-Augmented Transformer for Sequence Modeling
> Qingyang Wu, Zhenzhong Lan, Kun Qian, Jing Gu, Alborz Geramifard, Zhou Yu
>
> [2] Associative Recurrent Memory Transformer
> Ivan Rodkin, Yuri Kuratov, Aydar Bulatov, Mikhail Burtsev
>
> [3] Recurrent Memory Transformer
> Aydar Bulatov, Yuri Kuratov, Mikhail S. Burtsev

---

> > ### Author Response · Authors · 2025-11-26
> >
> > Thank you for these detailed suggestions, we agree that these baselines will be great to add to the paper. We will run experiments with the Recurrent Memory Transformer (RMT) architecture and add the results to the manuscript.
> > Note that in our theoretical section, we consider a generalization of SSMs, which includes any model that has a fixed memory as a function of the sequence length, and therefore RMT satisfies this property as well.

---

### Official Review · Reviewer_5b4b · 2025-10-26

**Soundness:** 3
**Presentation:** 2
**Contribution:** 3
**Rating:** 4
**Confidence:** 2

**Summary:**

This paper proves that SSMs cannot solve long-form generation tasks due to their fixed-size memory, but that interactive tool-use enables them to achieve length generalization—allowing SSMs to handle arbitrarily long and complex problems by leveraging external memory.
Empirically, the authors show that tool-augmented SSMs (e.g., Mamba) outperform Transformers in arithmetic, reasoning, and coding tasks, demonstrating scalability and efficiency advantages when used as interactive agents rather than standalone models.

**Strengths:**

- The paper proves that SSMs, despite their efficiency, cannot solve long-form generation tasks under standard or single-turn settings.
- The paper validates the theory on diverse task categories, including arithmetic, algorithmic reasoning, and real-world coding.

**Weaknesses:**

- The experiments primarily use small or mid-sized models. These experiments cannot prove if the proposed methods could generalize to larger scales.
- The paper evaluates on synthetic arithmetic, algorithmic, and coding tasks only. There is no testing on natural-language, multimodal, or real reasoning benchmarks.
- Although tool-use extends memory, it introduces I/O and latency overhead, which may offset the computational gains of SSMs in real-time systems.
- Typos, such as "node node" in line 398

**Questions:**

- How this proposed methods could generalize to real-world reasoning or language understanding tasks?

---

> ### Author Response · Authors · 2025-11-20
>
> We thank the reviewer for their great feedback, and provide a response below.
>
> Weaknesses:
>
> 1. Large-scale models: When training models from scratch, we limited experiments to relatively small models so that they can be easily reproduced without significant computational burden. We also experimented with pretrained 1.4B SSM and Transformer models, which show trends that are similar to what we observe for small models. We believe that we will see similar behavior in larger scale models. Indeed, different works [1,2] have shown that even frontier Transformer-based models struggle with tasks that involve long-context or long-form generation. Unfortunately, there are currently no frontier-level SSM-based model that we can compare these models against, and training one is beyond the scope of this work.
> 2. Evaluation on natural-language benchmarks: we acknowledge that our work lacks a natural-language task. Following this suggestion, we ran additional experiments on a natural-language task which tests long-context capabilities, using one of the tasks suggested in the recent Oolong benchmark [2]. This task involves computing some “statistics” over a dataset provided in-context. In this experiment we finetune a Pythia Transformer and a Mamba SSM (similar to the coding task), where we train on trajectories that invoke a simple tool interaction and chain-of-thought. Similarly to other experiments in the paper, here too we observe that while both models perform similarly when evaluated in-distribution, the Mamba SSM extrapolates better beyond the length of the training data (Figure 3 in the updated manuscript). We updated the paper with these additional results discussed in Section 3.4. We believe these new results strengthen the message of our paper, and thank the reviewer for this suggestion.
> 3. Latency of tool-use: we agree that tool-use can introduce latency due to various overheads, for example if the tool-use involves querying a remote server. However, in many cases the tool-use can be implemented efficiently, and the bottleneck is often the model inference, which involves massive compute over expensive hardware (unlike tools, which are typically implemented on cheap CPUs). We will therefore add a discussion on this point, acknowledging that the benefit of SSMs with tool-use depends on the exact setting, environment and solution engineering.
> 4. We will fix the typo in the paper.
>
> Questions:
>
> 1. Generalizing to real-world language understanding: see above, we add a natural language understanding task and observe similar benefits of using SSMs over Transformers.
>
> We believe we answered the concerns raised in the review. In particular, we think that our new experiment on the natural-language task, which we ran following your suggestion, provides a clear answer to the main concern raised in the review. If this is indeed the case, please consider raising your score, or otherwise please let us know if there are still any concerns that we might address.
>
> [1] Shojaee et al., 2025. The Illusion of Thinking: Understanding the Strengths and Limitations of Reasoning Models
> via the Lens of Problem Complexity.
> [2] Bertsch et al., 2025. Oolong: Evaluating Long Context Reasoning and Aggregation Capabilities

---

> ### Comment · Reviewer_5b4b · 2025-11-27
>
> Thanks for the author's detailed responses! I raised my score to 6 accordingly! But I cannot find ways to edit my score at this time.

---

### Official Review · Reviewer_14Xt · 2025-10-30

**Soundness:** 3
**Presentation:** 3
**Contribution:** 3
**Rating:** 8
**Confidence:** 4

**Summary:**

This paper argues theoretically and empirically for augmenting SSM architectures with tools that can function as an external memory. In principle, this provides a simple way to overcome the memory limitations of recurrent architectures (as opposed to the more conservative approach of hybridizing with attention layers). The authors first formalize the value of memory tools with some basic theory and then show empirically that SSMs using external memory tools can achieve good performance and length generalization on synthetic tasks. They also show an encouraging proof-of-concept result where (distilled) SSM coding agents with tool use better generalize to large codebases than transformer coding agents.

**Strengths:**

1. The overall argument of the paper is well motivated and presented. I appreciate the principled framing even if the theory is a bit surface-level
2. The question of how to extend SSMs with additional memory is very timely, and the community has not converged on a solution. One approach (e.g., used by Qwen-3-Next) is to hybridize by mixing transformer and SSM layers. But this paper proposes and evaluates a more radical solution without attention, which is valuable.
3. The empirical results in the paper are quite interesting. In particular, the empirical results showing better length generalization for distilled SSM coding agents with tool use (vs. transformer agents) is a promising proof of concept that the synthetic-task results are practically relevant

**Weaknesses:**

### Long-Form Task Definition

A task is defined to be long-form generation if the output size (number of possible outputs with probability mass) grows with input size n. I wouldn't necessarily agree with this definition. What if there is only one choice for y given x, but that y has to be large, and it's length scales with y - why don't we consider this long-form generation?

### Theory Reinvents the (Automata-Theoretic) Wheel

The theoretical results are somewhat superficial. While I appreciate the principled framing, they essentially boil down to an argument that, if a model has fixed memory, tools that serve as interface to additional memory are helpful for memory-intensive tasks.

> Note that any model that has fixed memory as a function of the sequence length satisfies the definition of a GSSM.

Your definition of a generalized state-space model with constant |S| is just reinventing the notion of a probabilistic finite automaton (with a potential distinction based on the long-form task definition thing mentioned above, though I'm not sure it's a meaningful one). See, for example, [Merrill et al., Section 3](https://arxiv.org/abs/2404.08819) for a discussion of the connection between SSMs, state tracking tasks, and automata/regular languages.

In formal language theory terms, your notion of GSSMs with interactive tool use has a natural interpretation as finite automata augmented with some tool data structure, similar to how pushdown automata are automata augmented with a stack data structure (and this lets pushdown automata solve additional languages that require more memory). Interaction is essentially giving the model more memory through interaction with an external data structure. I think making the connection between GSSMs and automata explicit and discussing this interpretation of tools as data structures would help clarify your argument.

Theorem 2.1 is just the Myhill-Nerode theorem (a probabilistic extension of the basic deterministic one)

Theorem 2.2: it's a bit overly verbose to talk about learning algorithms when this is essentially an expressivity construction. i.e., the learning algorithm and training data you have here is basically just hard-constructing an expressivity construction?

**Questions:**

Table 1: clarify this is with tool use. To really make your point that tools are necessary, you could show performance without tools.

what does it mean to move a pointer left or right? Increment/decrement it?

It seems your "logical graph problem" is really the circuit evaluation problem, which is known to be P-complete and thus outside the representational power of transformers unless the complexity classes TC0 and P collapse. Furthermore, even with CoT, it would be very surprising if transformers could solve this problem with a small number of steps (e.g., even with log^k n steps, CoT transformers cannot solve this problem unless TC0 and P collapse).

Do you have opinions on whether SFT alone would suffice to encourage tool use with SSM models? Or would RL be helpful for teaching a model to acquire tool use?

The following statements at the abstract level could be made more clear, since it's not sure what long-form generation or tractable mean at this point:

> We begin this work by showing a simple theoretical result stating that SSMs cannot accurately solve any long-form generation problem, undermining their main competitive advantage.

> SSMs can learn to solve any tractable problem and generalize to arbitrary problem length/complexity

Regarding length generalization with SSMs:
> SSMs have been shown to display robust length generalization capabilities in certain cases.

IMO you could make a stronger claim. It seems quite robust at this point that SSMs show strong length generalization relative to transformers. e.g., the gated delta net, path attention, and samba papers show zero-shot results consistently outperforming transformers on perplexity evals and RULER.

---

> ### Author Response · Authors · 2025-11-20
>
> We thank the reviewer for the very positive feedback on our paper and for their helpful comments. Below is a response to the different weaknesses and questions raised by the reviewer:
>
> Weaknesses:
>
> 1. Definition of Long-form generation: In our definition of long-form generation, we require that the (effective) number of outputs, over all possible inputs, grows to infinity. We do not require that the number of outputs for any individual input grows, and the case where each input has a single output is thus also considered long-form generation, as long as the total number of outputs grows (which will be the case if the length of the output is growing with the input). We will clarify this in the text, and please let us know if there is any confusion regarding this.
> 2. Theory reinventing the wheel: We completely agree that there is some overlap between our theoretical results and known results in automata theory, and we will add an explicit discussion about this in the main text. However, while the connection between SSMs and regular languages has been discussed in previous works, as you pointed out, our framework which studies SSMs with tool-use and their capabilities for solving long-form generation problems is, to our knowledge, novel.
>     - Theorem 2.1 - We agree that this result can be proved using a probabilistic version of Myhill-Nerode, and we will mention this when discussing the results. For completeness, we give a full proof that does not rely on prior results from automata theory.
>     - Theorem 2.2 - We wanted to discuss learning, and not just expressivity, so that we can study length generalization (extrapolating beyond the training distribution length). We agree that the result is not very far from an expressivity construction, as our training data and algorithm makes learning almost “trivial”, but it does allow us to discuss learning with finite sample complexity and with a finite (training) sequence length. That said, we believe that a similar result can be shown for more natural algorithms (e.g., training a simple SSM with gradient-descent), but this will significantly complicate the analysis, and we leave this to future work.
>
> Questions:
>
> 1. Table 1 is indeed with tool-use, and we will clarify this in the text. We ran additional ablations (detailed in Appendix G), training models in different settings without tool-use, and observe little to no length generalization. This observation aligns with the main argument in the paper, about the necessity of tool-use for solving long-form tasks.
> 2. Moving pointers: each pointer points to an “index” in the context, and moving left or right means incrementing/decrementing this index. When a “read” command is issued to the pointer, it retrieves the token in this index.
> 3. Logical graph problem: this problem is indeed very similar to the circuit evaluation problem. We note that a Transformer can solve this problem using CoT of linear size (in the circuit description) by evaluating the output of every gate.
> 4. We restrict our experiments to SFT in order to keep the experimental setting simple and reduce computational burden. However, we believe that exploring RL for encouraging tool-use in SSMs is an exciting future direction, and we believe that it will be crucial for developing frontier SSM-based tool-calling models.
> 5. We will improve the additional unclear statements, as suggested by the reviewer.

---

### Official Review · Reviewer_ubi8 · 2025-10-31

**Soundness:** 3
**Presentation:** 3
**Contribution:** 3
**Rating:** 8
**Confidence:** 3

**Summary:**

This paper investigates the length generalization capabilities of State Space Models (SSMs) in long-form generation tasks. The authors prove that SSMs cannot solve such tasks with bounded memory alone, but can achieve length generalization when augmented with interactive tool use. Experiments on arithmetic, reasoning, and coding tasks demonstrate that tool-augmented SSMs can generalize from short training sequences to much longer test sequences, outperforming Transformers in certain settings.

**Strengths:**

1. The paper provides formal definitions (GSSM, long-form generation tasks) and proves a crisp dichotomy: SSMs fail without interaction but succeed with interactive tool access.

2. The arithmetic experiments show remarkable generalization.

3. The experimental coverage is quite comprehensive. The paper evaluates multiple architectures (Mamba, LSTM, GRU, Pythia, Mistral) across diverse tasks (arithmetic, logic, coding), with both synthetic and agent-generated trajectories.

4. Connecting SSM limitations to tool-augmented agents is timely and relevant, given the growing interest in agentic AI systems.

5. The progression from theory to synthetic experiments to realistic coding tasks is logical and easy to follow.

**Weaknesses:**

1. The theory relies on training with complete Turing machine traces, but the experiments use hand-crafted or distilled trajectories with no analysis of this gap.

2. The key coding experiment uses a synthetic benchmark with a single bug pattern, not a complex real-world benchmark (e.g., SWE-bench).

3. Missing Sample Complexity Analysis: The theory proves learnability (Lemma B.1) but gives no practical bounds on the sample size (m) or task complexity (n_0) required to achieve it.

4. Incomplete baselines: The paper omits comparisons to SOTA hybrid SSM-Attention models (e.g., Jamba), which are a key alternative for efficient long-context modeling.

**Questions:**

1. How do the hand-crafted trajectories in arithmetic experiments satisfy (or approximate) the Turing machine trace assumption in Theorem 2.2?

2. In the coding experiments, you filter for successful trajectories. How does this affect the model's ability to handle errors or suboptimal tool use at test time?

3. Have you tried evaluation on actual software engineering tasks (e.g., SWE-bench lite)? What performance would you expect?

---

> ### Author Response · Authors · 2025-11-20
>
> We thank the reviewer for the very positive feedback on our paper and for their helpful comments. Below is a response to the different weaknesses and questions raised by the reviewer.
>
> Weaknesses:
>
> 1. Mismatch between using complete Turing machine traces and experiment trajectories: in order to get a very general theoretical result that applies to any computable function, we can only use the Turing-machine computability property. Therefore, the only information we can leverage is the Turing-machine traces, which can be overly complicated and “verbose” for practical problem. To make training and evaluation more efficient, in the experiments we use trajectories that capture the relevant computation but require shorter sequences than the formal Turing machine implementation. We believe that this still captures the “spirit” of the theoretical analysis.
> 2. Simplified coding experiment: our goal in the coding problem setup is to generate coding problems where we can exactly control the complexity of the problem. This would be hard to do for more “natural” coding problems such as the ones in SWE-bench. Moreover, since there are currently no frontier SSM-based coding agents, solving complex coding problems in real SWE benchmarks with SSMs requires significant effort that goes beyond the scope of this paper. However, since we collect trajectories from a real-world coding agent, the resulting trajectories are more “natural” than the synthetic experiments.
> 3. Missing sample complexity: in the proof of Lemma B.1 we derive the sample complexity bound, which depends on the value of n_0 as well as the size of the Turing machine. We will add this sample complexity to the statement of the result. As for the task complexity n_0, this value depends on the operation of the underlying Turing machine, and is defined and discussed in the Lemma.
> 4. Hybrid-SSM baseline: this is a great suggestion. We will run additional experiments with hybrid-SSMs and add them to the paper.
>
> Questions:
>
> 1. Turing machine traces: see answer 1 above.
> 2. Filtering positive trajectories: we see overall that filtering for correct trajectories improves performance. Note that these trajectories can still contain errors or suboptimal tool-use, but in cases where the teacher model was able to recover from the errors and complete the task. Hence, this filtering in fact can improve the ability of the model to handle errors. This filtering can be seen as a single step of Expert Iteration or STaR (a naive form of Reinforcement Learning), which has been reported to work well in different settings. [1]
> 3. Evaluation on software engineering tasks: as we mention above, there are currently no SSM-based coding agents that can handle real SWE benchmarks, and training one requires significant effort and is beyond the scope of this work. We believe that if such an agent was trained, it would perform better on long-horizon coding problems, as suggested by our experiments.
>
> [1] Zelikman et al., 2022. Star: Bootstrapping reasoning with reasoning

---

### Author Response · Authors · 2025-12-02

Dear AC,

We spent significant effort in addressing the criticism raised by the reviewers. In particular, we ran additional experiments following the suggestions of Reviewer 5b4b. Our experiments show that our observations also hold for a natural language reasoning task which we describe in an updated revision of the paper. Following our revision, Reviewer 5b4b has agreed to raise their score from 4 to 6.

We wish to thank again the reviewers for the overall very positive and helpful comments they provided, and hope that our effort in responding to the reviews will be considered when the decision is finalized.

Best,
Authors

---

### Meta-Review · Area_Chair_9rLE · 2025-12-23

**Summary:**

This paper investigates the length generalization capabilities of State Space Models (SSMs) compared to Transformers. The authors establish a theoretical dichotomy: while SSMs with fixed memory cannot solve long-form generation tasks, augmenting them with interactive tool-use (essentially an external memory tape) allows them to solve any tractable problem and generalize to arbitrary lengths. Empirically, the paper demonstrates that Mamba models trained with tool-use trajectories (e.g., pointers, bash commands) achieve almost perfect length generalization on arithmetic, logical reasoning, and coding tasks, significantly outperforming Transformer baselines (Pythia, Mistral) which were trained in the exact same tool-use setting but failed to extrapolate.

The reviewers were largely enthusiastic. Initial concerns focused on the reliance on synthetic tasks, the lack of comparisons to memory-augmented architectures (like RMT), and the overlap between the theoretical proofs and existing automata theory. The authors provided a well-formed rebuttal, notably adding a natural language experiment (Oolong benchmark) and agreeing to add missing baselines, which solidified the consensus for acceptance  among the reviewers. The AC agrees with the consensus and recommends Accept, but urge the authors to add baselines (e.g. RMT and Hybrid-SSM) in the final version as promised in the rebuttal.

**Reviewer Concerns:**

Addressed Concerns
- Lack of Non-Synthetic Tasks: Reviewer 5b4b initially critiqued the paper for relying solely on synthetic arithmetic and coding tasks. The authors addressed this by adding a long-context natural language experiment adapted from the Oolong benchmark. The results showed that SSMs maintained performance on longer sequences where Transformers degraded, leading the reviewer to raise their score.
- Missing Baselines: Reviewers ubi8 and Mthj noted the absence of comparisons to Hybrid SSM-Attention models and memory-augmented Transformers (e.g., Recurrent Memory Transformer), which are stronger competitors than standard Transformers. The authors agreed to run and include these experiments (RMT and Hybrid-SSM) in the final version.
- Sample Complexity: Reviewer ubi8 noted the theory proved learnability but lacked sample complexity bounds. The authors addressed this by explicitly deriving the sample complexity in Lemma B.1 and promising to add it to the main theorem statement.

Outstanding Concerns
- Theoretical Novelty: Reviewer 14Xt argued that the theoretical contributions essentially "reinvent the wheel" of automata theory (specifically the Myhill-Nerode theorem) and that "tools" are simply external data structures for automata. While the authors acknowledged this connection and promised to cite relevant literature, the core critique that the mathematical contribution is an application of known complexity theory rather than a new derivation remains valid, though the application to modern SSM agents is novel.
- Gap Between Theory and Experiments: Reviewer ubi8 pointed out that the theory relies on full Turing machine traces, while experiments use distilled or hand-crafted trajectories. While the authors justified this as a necessary efficiency trade-off, the formal gap remains an outstanding limitation of the analysis.

**Reviewer Scores:**

- Reviewer ubi8: (8 to 8). The reviewer was already confident and positive. The clarification on sample complexity and the promise of hybrid baselines likely reinforced this score.
- Reviewer 14Xt: (8 to 8). Despite the theoretical critique regarding automata theory, this reviewer found the framing "principled" and the empirical results "interesting," maintaining a high score throughout.
- Reviewer 5b4b: (4 to 6). This reviewer explicitly raised their score from 4 to 6 during the discussion period after the authors added the natural language experiment.
- Reviewer Mthj: (8 to 8). The reviewer was "absolutely certain" in their assessment and praised the work as "excellent." The inclusion of the RMT baseline requested would likely solidify this high rating.

---

### Decision · Program_Chairs · 2026-01-26

Accept (Oral)